# Multi-stimuli-responsive programmable biomimetic actuator

Yue Dong[1,4], Jie Wang[1,2,4], Xukui Guo[1], Shanshan Yang[1], Mehmet Ozgun Ozen[2], Peng Chen[1], Xin Liu[1], Wei Du[1], Fei Xiao[3], Utkan Demirci ⓘ [2] & Bi-Feng Liu ⓘ [1]

Untethered small actuators have various applications in multiple fields. However, existing small-scale actuators are very limited in their intractability with their surroundings, respond to only a single type of stimulus and are unable to achieve programmable structural changes under different stimuli. Here, we present a multiresponsive patternable actuator that can respond to humidity, temperature and light, via programmable structural changes. This capability is uniquely achieved by a fast and facile method that was used to fabricate a smart actuator with precise patterning on a graphene oxide film by hydrogel microstamping. The programmable actuator can mimic the claw of a hawk to grab a block, crawl like an inchworm, and twine around and grab the rachis of a flower based on their geometry. Similar to the large- and small-scale robots that are used to study locomotion mechanics, these small-scale actuators can be employed to study movement and biological and living organisms.

[1] Britton Chance Center for Biomedical Photonics at Wuhan National Laboratory for Optoelectronics-Hubei Bioinformatics & Molecular Imaging Key Laboratory Systems Biology Theme, Department of Biomedical Engineering, College of Life Science and Technology, Huazhong University of Science and Technology, Wuhan 430074, China. [2] Bio-Acoustic MEMS in Medicine (BAMM) Laboratory, Canary Center at Stanford for Cancer Early Detection, Department of Radiology School of Medicine Stanford University, Palo Alto, CA 94304, USA. [3] School of Chemistry & Chemical Engineering, Huazhong University of Science and Technology, Wuhan 430074, China. [4]These authors contributed equally: Yue Dong, Jie Wang. Correspondence and requests for materials should be addressed to U.D. (email: utkan@stanford.edu) or to B.-F.L. (email: bfliu@mail.hust.edu.cn)

In recent years, smart materials, particular responsive materials such as actuators, have attracted great attention[1–13]. Actuators, which can be defined as machines that can convert external stimuli such as magnetic/electric fields[14–16], pH[17], temperature[18–21], solvent[22,23], humidity[24–27], and light[28,29] to mechanical movements, hold great potential in many frontier applications, including soft robots[30,31], microswimmers[32], tissue engineering[33], artificial muscles[34,35], electronic skins[36], target capture/release and biomimetic actuations[1,25]. Actuators with fast, controllable/programmable structure changes/shape transformations are the key factor for the initial design to achieve different applications. Several methods have been proposed to fabricate actuators with desired structural changes, which can be classified into two main types. In one type, which is inspired by fibrous plant organs, aligned and rigid matrices are embedded in a soft and pliant matrix to mimic the life behaviors of plant organs[37–41]. Hydrogel stripes have been incorporated into the network of certain hydrogels with different moduli by lithography to fabricate hydrogel actuators, and these hydrogel actuators can change from planar to cylindrical and conical helical structures[41,42]. Stiff reinforcing magnetic nanoparticles or platelets (iron oxide nanoparticles and functional ceramic platelets) with aligned orientation that can be controlled by an external magnetic field are introduced into networks of hydrogels and ceramics to produce hydrogel-based and ceramic-based actuators with programmable structural changes[38,40]. Plant-inspired composite hydrogel architectures similar to orchids and calla lily flowers are fabricated by four-dimensional (4D) printing, and these hydrogels are encoded with localized, anisotropic swelling behavior controlled by the alignment of cellulose fibrils along prescribed 4D printing pathways[1]. The other type is actuators with bilayer or multilayer structures, which are widely researched and consist of a single component with different orientations of different layers or several components in different layers, including an active layer that contracts or expands under external stimulation and a passive layer that remains intact[43–45]. Electrospinning has been employed to produce thermoresponsive bilayer hydrogel fibrous membranes and hydrogel fibers with different orientations at every layer showing reversible coiling, rolling, bending, and twisting deformations[19–21]. Lithography and direct laser writing (DLW) belong to the scope of light-mediated manufacture[46] and are very commonly used for the fabrication of actuators with layered structures, which are amenable to photoinitiation reactions, allow designable patterning and create a wide range of proof-of-concept devices, including various self-folding, origami and other complex deformable structures[47]. In total, photoinduced reactions are the only strategy for fabricating actuators that can achieve programmable structure changes, particularly actuators with bilayer structures.

Graphene and its derivatives are outstanding candidates for the fabrication of smart and high-performance bilayer actuators due to their fascinating properties, including extraordinary flexibility, low weight density, high mechanical strength and electrical conductivity, unique thermal and optical properties, and good stability[48–50]. In particular, graphene oxide (GO), as an important graphene derivative, has been chosen as building blocks for the fabrication of GO-based actuators[18,24–26,28,51–55]. A solo GO film-based actuator has proven the ultrafast response and large deformation degree due to the anisotropic structures (called Quantum confined superfluidics channels), and distinct advantage in long-term stability owing to the single composition[56]. GO fiber-type and GO film-type smart actuators with adjustable structural changes are successfully prepared by DLW on GO fibers and GO films along a predesigned path[18,51]. A simple laser-scribing procedure also has been used to fabricate GO-based programmable actuators[57]. In the process of DLW and laser-scribing, GO is reduced to reduced GO (RGO), so RGO/GO bilayer structures form on a specific zone of the GO fiber and GO film, which is vital for the formation of actuators due to the apparent difference in expansion and shrinkage between RGO and GO upon adsorption/desorption of water molecules. RGO/GO bilayer actuators with programmable structural changes have also been prepared by UV- and sunlight-irradiation-induced photoreduction of GO films, in which photoreduction of a specific zone is achieved with the assistance of a photomask[25,26]. Therefore, GO-based actuators with programmable structural changes can also be produced by only photoinduced reactions owing to their capability of precise local modification. However, apart from photoinduced reactions, local modifications with designable patterning have not been reported, thereby restricting the development of some possible actuators with potentially excellent performances, for example, GO/polymer bilayer actuators. As previously reported literature[58,59], high-performance actuators have been fabricated by modifying the PDA polymer onto the GO sheet, but the direct local modification of polymer layer onto the GO film still cannot be achieved.

In this paper we report a simple but efficient and robust method for the fabrication of a programmable GO/polymer bilayer actuator via a precise local reaction with the assistance of a solvent-loaded agarose hydrogel stamp. We chose polypyrrole (PPy) as a representative polymer owing to its outstanding performance in previously reported actuator systems[24,27,52,53]. Interestingly, this programmable GO/PPy bilayer actuator could be fabricated quickly and exhibited excellent actuating performance under the stimuli of humidity, temperature and infrared (IR) light due to the adsorption/desorption of water molecules. We also achieved local modification of polyaniline (PANI), polyethylene dioxythiophene (PEDOT) and calcium alginate hydrogel on GO film, and PPy on GO fibers and GO foam. As representative examples, GO/PPy bilayer actuators that mimic a hawk claw and tendril climber plants were designed for controllable object transport. Furthermore, we also mimicked the movement of the inchworm, which paves the way for the development of soft walking robots.

## Results

**Fabrication strategy and programmable structure changes.** Figure 1a shows the fabrication process of programmable GO/PPy bilayer actuators. Similar to soft lithography, agarose hydrogel stamps with specific patterns are replicated from a polydimethylsiloxane (PDMS) mold[60], then peeled off and soaked in $FeCl_3$-containing aqueous solution. Meanwhile, GO film is fabricated by dropping GO suspension onto a flat plate and drying at ambient temperature. A $FeCl_3$-loaded agarose hydrogel stamp is placed onto the GO film, the microcontact patterning process is continued for 20 s, and $FeCl_3$ is transferred onto the GO film due to the adsorption of $FeCl_3$ aqueous solution by GO, resulting in the formation of a thin aqueous film of $FeCl_3$ with specific patterns on the contact area. As shown in Supplementary Fig. 1, the color of these specific patterns changed from golden yellow to brown. The $FeCl_3$-loaded agarose hydrogel stamp can be reused after soaking in the $FeCl_3$ solution once again because no damage occurs in the process of microcontact patterning. Upon removal of the $FeCl_3$-loaded agarose hydrogel stamp, pyrrole monomer is dropped onto the GO film, and the reaction between $FeCl_3$ and the pyrrole monomer occurs immediately to form PPy patterns on the GO film, leading to a color change in the specific patterns from brown to black, whereas no change can be observed in the region without $FeCl_3$. Figure 1b displays the optical image of PPy patterns with different sizes and shapes on the GO film. We can see clearly that black PPy patterns are

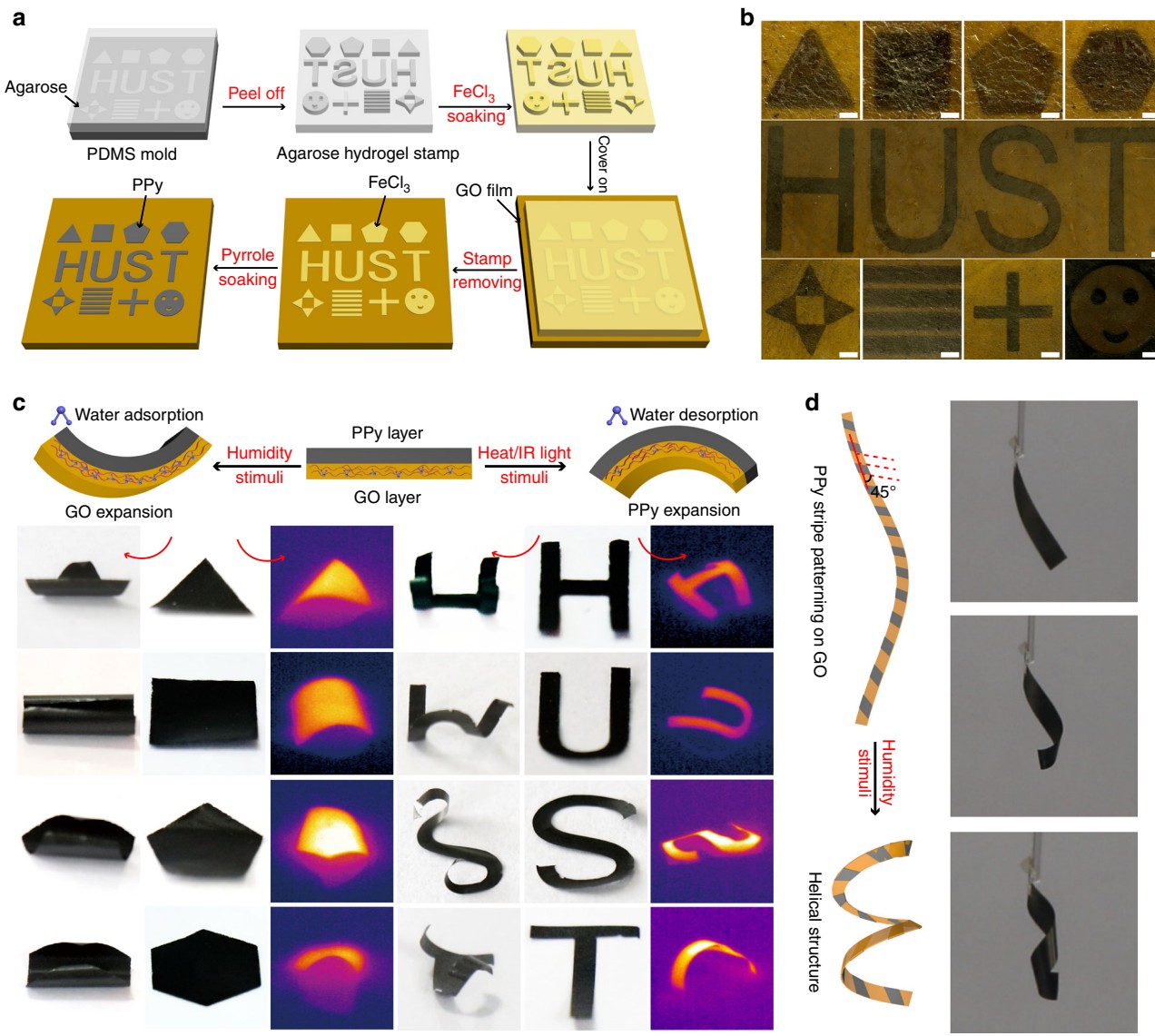

**Fig. 1** Fabrication and programmable structure changes of actuator. **a** Schematic diagram of precise PPy patterning on a GO film for the fabrication of a programmed GO/PPy actuator; **b** optical image of precise PPy patterning on a GO film (Scale bar: 100 μm); **c** working mechanism of GO/PPy bilayer actuator and the structural changes of actuators with regular triangle, square, regular pentagon, regular hexagon, and H-, U-, S-, and T-shape patterns under the stimuli of humidity and IR light; **d** schematic diagram of a programmable GO/PPy actuator

introduced onto the GO film with high precision. These patterns are designed for different functions, including triangle, rectangle, pentagon, hexagon, cross, stripe array, smiling face, and H-, U-, S-, and T-shapes, as well as other complex patterns. We demonstrate the high precision of PPy patterning by calculating the average width difference between the PDMS mold (100, 200 and 500 μm in width) and corresponding PPy stripes ($n = 10$). As shown in Supplementary Fig. 2, the average differences are 6.7, 6.5 and 8.1 μm, respectively, which, in comparison to the bulk of actuators, can be ignored. The highest resolution of this method can reach to 60 μm (Supplementary Fig. 3). Similarly, we achieve precise modification of PANI and PEDOT patterns by changing the pyrrole into the aniline and EDOT, and calcium alginate hydrogel patterns are introduced onto GO film with high precision by changing FeCl₃-loaded agarose hydrogel stamp and pyrrole into CaCl₂-loaded agarose hydrogel stamp and sodium alginate, respectively (Supplementary Fig. 4 and Supplementary Note 1, Supplementary Fig. 5 and Supplementary Note 2, Supplementary Fig. 6 and Supplementary Note 3). Our presented

method is also applied to the GO fiber and foam. Similarly, black PPy patterns can also be modified onto GO fibers and foam, which indicates the possibility for the development of GO fiber- and foam-based actuators (Supplementary Figs. 7 and 8). The microcontact time between the GO film and FeCl₃-loaded agarose hydrogel stamp, which influences the amount of FeCl₃ adsorption and PPy modification, is important for the fabrication of GO/PPy bilayer actuators. As shown in Supplementary Fig. 9, the color of the PPy pattern darkens gradually with increasing microcontact time, verifying that an increasing amount of PPy is patterned onto the GO film. However, the color did not change when the microcontact time is longer than 20 s. We speculate that enough FeCl₃ has been transferred onto the GO film within 20 s to form a compact and dense PPy film, which blocks the diffusion of the pyrrole monomer; therefore, the reaction between FeCl₃ and the pyrrole monomer is restricted, and PPy cannot be produced continuously. Considering previously reported bilayer actuators, a compact and dense film is crucial for the formation of bilayer actuators, so the optimized microcontact time is important. The

fabricated GO/PPy bilayer actuator can work under the stimuli of humidity, temperature and IR light (diagram in Fig. 1c). The GO layer expands with increasing humidity due to water adsorption, whereas the PPy layer is almost inert to humidity changes, leading to the bending of the actuator relative to its PPy layer. IR light or heating treatment can increase the temperature of the actuator, which will make the GO layer contract owing to desorption of water molecules, while the PPy layer expands because of its inherent nature; thus, the GO/PPy bilayer actuator bends relative to its GO layer. As shown in Fig. 1c, an actuator with a specific pattern will bend and contract relative to the PPy layer in a different structure under the stimuli of humidity and IR light (Supplementary Figs. 10–25). Note that the bending and contraction of the actuator are always along the axis of symmetry when the actuator has a regular pattern, such as a regular triangle, square, regular pentagon or regular hexagon. The bending direction is mainly determined by the longer symmetrical axes and larger structure change of actuator will happen along the direction. However, a synergistic effect should be considered for the structural changes of actuators with special patterns, such as H-, U-, S-, and T-shape patterns. In particular, GO/PPy bilayer actuators with precise PPy patterning can achieve more complex programmable structure changes. As shown in Fig. 1d, the GO/PPy actuator with PPy diagonal stripes can transform into helical structures when stimulated by humidity.

**Structural characterization of GO/PPy actuator**. As shown in Fig. 2, the structural characteristics of the actuator were explored. Figure 2a shows a scanning electron microscope (SEM) image of the GO film side. Characteristic wrinkles[18], which are identical to previously reported results, can be seen clearly. Compared to a film with a flat surface, this wrinkled GO film can provide more sites and can ensure a close combination of the GO film and PPy film. In recent years, concerns have been focused on mechanical stabilities between different material layers, which is a major barrier to their practical applications with excessive use; thus, a close combination between the GO and PPy films is favorable for the mechanical stability of GO/PPy bilayer actuators. Furthermore, more sites can provide more area for the GO film to

contact water molecules, and the unimpeded permeation of water through the GO film has been demonstrated[61], so the GO film will be sensitive to changes in relative humidity (RH). The SEM image of the PPy side in Fig. 2b and the cross-sectional SEM images of PPy film in Fig. 2c–e reveal that the PPy film possesses dense structure, which is inconvenient for the transfer of water molecules in the network of PPy film. Furthermore, the hydrophilic functional group is lacking in the PPy film and the interaction between water molecules and PPy film is weak, and the polymeric structure of PPy also make it hydrophobic. Therefore, the PPy film is not sensitive to changes in RH. Static water contact angle (CA) measurements were applied to characterize the surface wetting of the two sides (Supplementary Fig. 26). The PPy side of the PPy film exhibits a significantly increased CA of ca. 73.8°, whereas the reverse side features a CA of ca. 52.2°, which is similar to that of a pure GO film. Generally, an obvious change in surface wettability could be mainly ascribed to the formation of a dense PPy film. Figure 2c–e display cross-sectional SEM images of GO/PPy bilayer actuators fabricated on the basis of GO films with different thicknesses of 12.8 μm, 25.2 μm and 37.1 μm. Strikingly, the thicknesses of the PPy films are equal and the thickness we measured from the cross-sectional SEM image of GO/PPy is about 15.2 μm. Therefore, the thicknesses of PPy films are not influenced by GO film, which makes it convenient to explore the correlation of GO film thickness with actuator performance.

X-ray photoelectron spectroscopy (XPS) was used to study the chemical status of C, N, and O elements in the GO/PPy actuator. As shown in Fig. 2f, the XPS spectra for the PPy side and reversible side of the GO/PPy actuator display a dominating C 1s peak ca. 286.1 eV and an O 1s peak ca. 532.2 eV, respectively. The C/O atomic ratio for the PPy side of the GO/PPy actuator is 3.04, which is slightly higher than that for the reversible side of the actuator (ca. 2.74), demonstrating the successful generation of PPy on the GO film. Furthermore, it has been reported that GO can be used as an oxidant to polymerize pyrrole into PPy, leading to the spontaneous partial reduction of GO after 3 days of interaction[24]. However, the reaction between $FeCl_3$ and pyrrole occurs immediately; therefore, the reduction of GO by pyrrole can be ignored and the C/O atomic ratio have nothing to do with

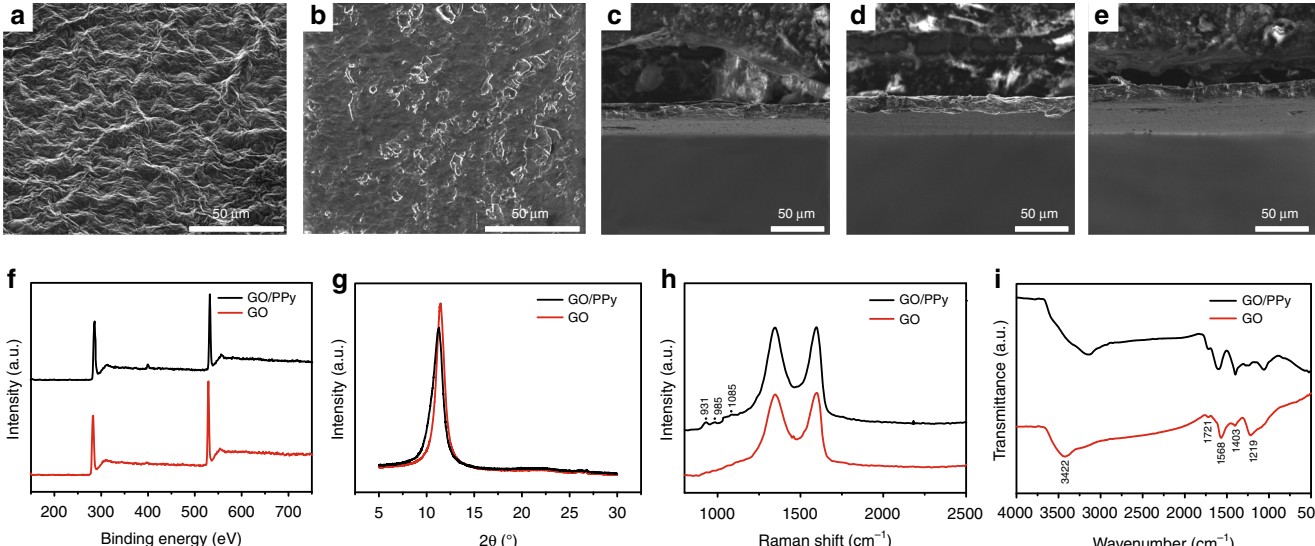

**Fig. 2** Structural characterization of GO/PPy actuator. SEM image of **a** the GO film side and **b** the PPy side; cross-sectional SEM image of GO/PPy actuators fabricated on the basis of GO films with different thicknesses of **c** 12.8 μm, **d** 25.2 μm and **e** 37.1 μm; **f** XPS spectra of GO and GO/PPy; **g** XRD pattern of GO and GO/PPy; **h** Raman spectra of GO and GO/PPy; **i** FT-IR spectra of pure GO and GO/PPy

the reduction of GO. An obvious N 1$s$ peak ca. 399.8 eV can be found in the XPS spectrum of the PPy side of actuator, demonstrating the presence of pyrrolic-like N atoms and successful incorporation of PPy into the GO film. X-ray diffraction (XRD) was conducted to further demonstrate the successful incorporation of PPy into the GO film. As shown in Fig. 2g, the GO film exhibits one sharp peak centered at 10.5°, corresponding to the (002) reflection of stacked GO sheets. However, the peak position of the PPy side of the actuator shifts from 10.5° to 10.2°, indicating larger interlayer spacing due to the embedding of PPy into the GO film. The surface compositions of the GO and PPy sides of the actuator are further characterized by Raman spectroscopy (Fig. 2h). GO displays two dominating peaks at 1350 and 1593 cm$^{-1}$, corresponding to the well-documented $D$ and $G$ bands, respectively. For the PPy side of actuator, a series of peaks at ca. 931, 985 and 1085 cm$^{-1}$ are consistent with those of polymerized PPy[62,63], further demonstrating the successful polymerization of pyrrole on the GO film. In addition, the $I_D$/$I_G$ of the GO side is 0.977, which is close to the $I_D/I_G$ value f (0.997) of the PPy side, revealing that no apparent defects are introduced into the GO film during the polymerization of pyrrole. Fourier transform infrared spectroscopy (FT-IR) spectra of the GO and PPy sides of the actuator are presented in Fig. 2i. The characteristic IR peaks of pristine GO agree well with those in the literature[64,65], including a broad band at 3422 cm$^{-1}$ corresponding to the O–H stretching vibration, a band at 1568 cm$^{-1}$ assigned to the C = C stretching vibration, a band at 1721 cm$^{-1}$ corresponding to the carboxyl group, a band at 1219 cm$^{-1}$ signifying the C-OH stretching vibration, and a band at 1403 cm$^{-1}$ assigned to the carboxy C-O stretching vibration. For the PPy side of the actuator, several new peaks ca. 1275 and 1061 cm$^{-1}$ can be observed, which are attributed to C–H and N–H deformation vibrations, respectively. It is, therefore, evident that PPy is indeed loaded onto the surface of the GO film via interactions such as hydrogen banding and π–π stacking between PPy and GO.

**Actuating performance of GO/PPy actuator.** The actuating performance of GO/PPy actuators with a typical ribbon shape (width 2 mm and length 15 mm) was investigated quantitatively under the stimuli of humidity, temperature and IR light. As shown in Fig. 3a and Supplementary Movie 1, a rapid bending behavior from the GO side to the PPy side was observed due to the water-adsorption-induced expansion of the GO layer. In contrast, the PPy layer had negligible changes due to its very low water adsorption. GO/PPy (a), GO/PPy (b) and GO/PPy (c) represent GO/PPy actuators having GO layers with different thicknesses of 12.8 μm, 25.2 μm and 37.1 μm, respectively (Fig. 2c–e). Obviously, among these three structures, GO/PPy (b) exhibited the best actuating performance, and its curvature increased rapidly from 0 to 4.18 cm$^{-1}$ with increasing RH from 50 to 68%. The insets in Fig. 3a show photographs of the bending GO/PPy ribbon under different RH values. By contrast, the curvature of GO/PPy (a) increases from 0 to 3.09 cm$^{-1}$ with increasing RH from 50 to 70%, and the curvature of GO/PPy (c) increases from 0 to 2.28 cm$^{-1}$ within the same RH range. For GO/PPy (a), the GO layer is so thin that the water-adsorption-induced expansion force cannot bend the PPy layer quickly, so the curvature change is small in the given RH range. For GO/PPy (c), the large stiffness of the GO layer with a large thickness means that it cannot bend flexibly, so its curvature change is less than that of GO/PPy (b). For comparison, a pristine GO film with a thickness of 25.2 μm was also cut into ribbons of the same size, and no bending performance under different RH values is observed. The GO/PPy also has a response to humidity from 20

to 50%, and GO layer will shrink and the actuator bends to GO layer. As shown in Supplementary Fig. 27, the curvature of GO/ PPy (b) will change from 0 to −0.57 cm$^{-1}$ with decreasing humidity. The insets in Supplementary Fig. 27 show photographs of the bending GO/PPy (b) at humidity levels of 20.4%, 31.8%, 43.5% and 50%. The thickness of PPy layer will increase with the increasing contact time of FeCl$_3$-loaded agarose hydrogel stamp onto the GO film (Supplementary Fig. 9). As shown in Fig. 3b, the relationship between actuating performance and contact time is also investigated. The thickness of GO film used in the experiment is kept the same and is equal to the thickness of GO layer in GO/PPy (b), and the contact time of 10s, 15s and 20s are chosen, so the corresponding actuators are named as GO/PPy (d), GO/ PPy (e) and GO/PPy (b), respectively. Clearly, we can find that the GO/PPy (e) possesses the best actuating performance, which is slightly larger than GO/PPy (b). This can be explained by smaller stiffness for GO/PPy (e). The PPy thickness reduces with decreasing microcontact time, so the stiffness of actuator reduces synchronously. However, the GO/PPy (d) is smaller in actuating performance than GO/PPy (b) and GO/PPy (e), which is owing to the minor difference of water-adsorption ability between GO and PPy layer at the GO/PPy (d). The response speed can be used to compare the performance of actuators. We have systematically investigated the relationship between the response speed of the actuators and the configurations (thicknesses of two layers and their sizes). Supplementary Fig. 28 indicated the GO/PPy actuator with the GO thickness of 25.2 μm possesses larger response speed at the same microcontact time, and the GO/PPy (e) possesses maximum response speed, which is also slightly larger than GO/ PPy (b). Considering the increasing mechanical strength with the increasing of PPy thickness and the minor difference of actuating performance and response speed between GO/PPy (e) and GO/ PPy (b), so we choose GO/PPy (b) for further studies (Supplementary Note 4). Supplementary Fig. 29 exhibits the relationship between the response speed of GO/PPy and their sizes, we can find clearly that the response speed increases gradually with the increasing of actuator length at the same width, and the response speed decreases gradually with the increasing of actuator width, which is ascribed to the larger bending force for the larger actuator length and larger flexural rigidity for the larger actuator width[66]. Furthermore, GO/PPy (b) can recover its straight state rapidly when the RH decreases from 69 to 50%, so the bent and straight states of GO/PPy (b) can be switched between freely by changing the corresponding RH (Fig. 3c). The average response and recovery time for the bent and straight state are measured to be 9 and 18 s, respectively. GO/PPy (b) displays extraordinary stability during the continuous change of RH between 50 and 69%. We can see clearly from Fig. 3d that there is no obvious deterioration of the curvature after 100 cycles. GO/PPy (b) also exhibited excellent durable behavior, and no obvious decrease in curvature was observed after exposure to 57% or 67% RH for 600 s (Fig. 3e). The GO/PPy actuator can also work under the stimulus of temperature, which is the result of the combined action of the GO layer and PPy layer. The GO layer will shrink with increasing temperature due to water desorption, and a rapid bending behavior from the PPy side to the GO side is observed. Meanwhile, the PPy layer will expand and bend to the GO side with increasing temperature. As shown in Fig. 3f, the curvature will change from 0 to 3.43 cm$^{-1}$ with increasing temperature from 25 to 100 °C. The insets in Fig. 3f show photographs of the bending GO/PPy ribbon at temperatures of 25, 40, 50 and 100 °C. Notably, the curvature cannot be changed when the temperature increases from 100 to 120 °C. This behavior is due to minor changes in humidity and minor expansion of PPy in this temperature range. However, sharp expansion will occur when the temperature is larger than 120 °C, which makes the PPy

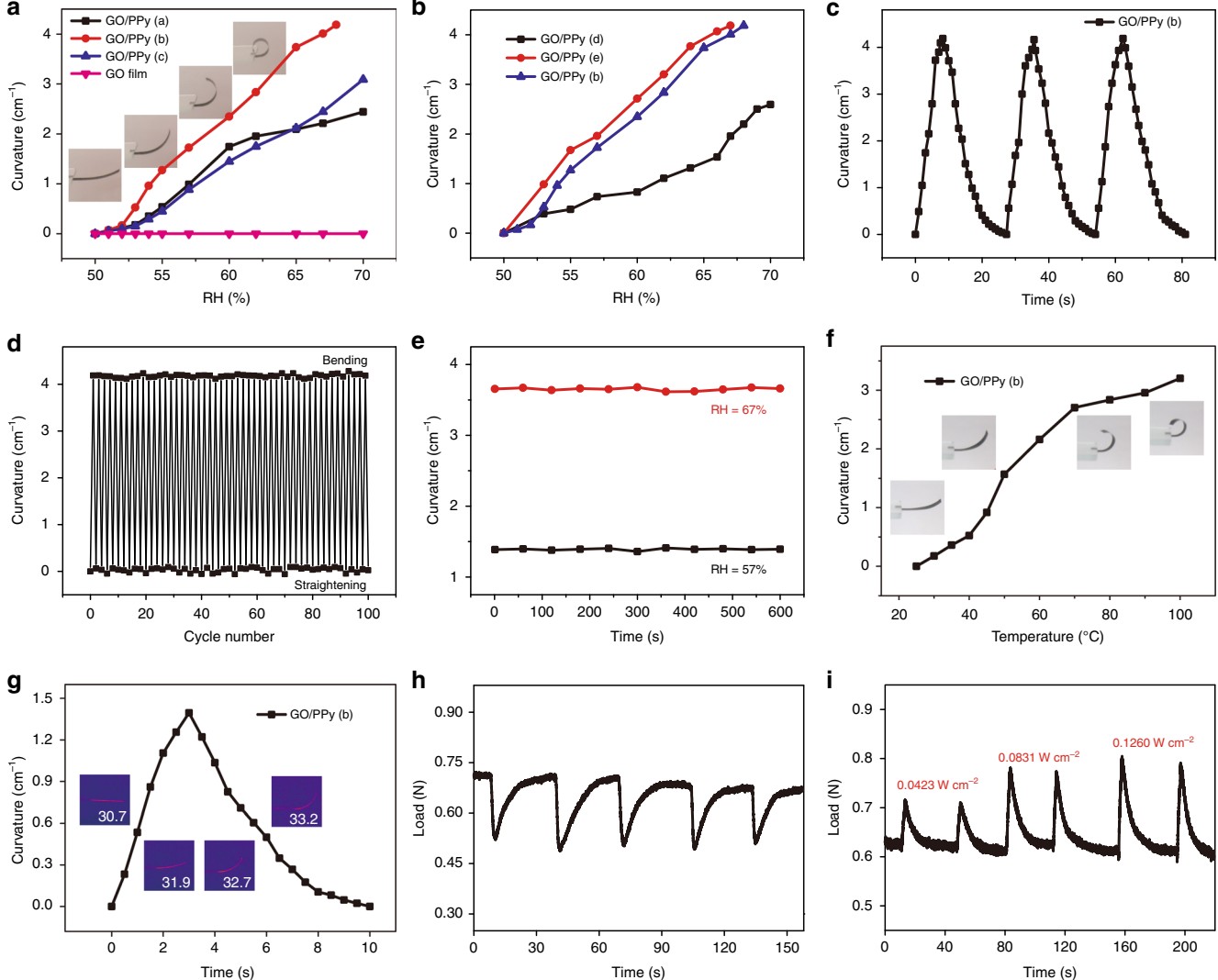

**Fig. 3** Actuating performance of GO/PPy actuator. **a** Actuating performance of GO/PPy actuators under the stimulus of humidity, (GO/PPy (**a**), GO/PPy (**b**) and GO/PPy (**c**) represent GO/PPy actuators with different GO thicknesses of 12.8, 25.2 and 37.1 μm, respectively) (the insets are photographs of the bending GO/PPy ribbon under different RH values); **b** Actuating performance of GO/PPy actuators under the stimulus of humidity, (GO/PPy (**d**), GO/PPy (**e**) and GO/PPy (**b**) represent GO/PPy actuators with increasing PPy thicknesses and same GO thickness of 25.2 μm; **c** switching from the bent to the straight state of GO/PPy (**b**) by changing the corresponding RH from 50 to 68%, and vice versa; **d** cyclic stability of the curvature of a GO/PPy actuator over 100 cycles (50–68% RH); **e** durable behavior of GO/PPy (**b**) after exposure to 57% or 67% RH for 600 s; **f** actuating performance of GO/PPy (**b**) under the stimulus of temperature (the insets are photographs of the bending GO/PPy ribbon at temperatures of 25, 50 and 100 °C); **g** actuating performance of GO/PPy (**b**) under the stimulus of IR light (the insets are photographs of the bending GO/PPy (**b**) at different times); **h** the bending force of GO/PPy (**b**) actuator recorded during the process of turning on/off the humidifier; **i** the contractile force of GO/PPy (**b**) actuator recorded during the process of turning on/off IR light with power densities of 0.0423, 0.0831 and 0.126 W cm⁻² (from left to right)

exfoliate from the GO layer; thus, the GO/PPy (b) actuator will be destroyed (Supplementary Fig. 30). GO/PPy (b) can also bend upon irradiation with IR light and its curvature can reach to 1.40 cm⁻¹ within 3 s (Fig. 3g and Supplementary Movie 2), and can recover its initial state within 7 s when the IR light is turned off. The mechanism of such responses is ascribed to the different degrees of water absorption/desorption by the GO and PPy layers. The water molecules will be released from the GO layer upon exposure to IR and be adsorbed by the GO film when the IR light is turned off. However, the PPy layer does not show such a photothermal response, so the GO/PPy (b) actuator will bend toward the GO side upon IR heating. The IR-induced temperature of the GO layer is approximately 33.0 °C (the inset in Fig. 3g), which is much lower than the decomposition temperature of GO; thus, the effect of the IR light on GO is negligible,

and the GO/PPy actuators display favorable cycling ability (Supplementary Fig. 31). The energy conversion efficiency of GO/PPy (b) under the stimulus of IR light with power densities of 0.0831 W cm⁻² is calculated to be 1.832%, which is higher than the efficiency of other reported actuator[13,67,68].The details of energy conversion efficiency calculation are described in the ESI (Supplementary Note 5). To accurately achieve the future applications of actuator, we also investigated the bending and contractile force under the stimuli of humidity and IR light, respectively. After increasing the RH by a humidifier, the force recorded by All-Electrodynamic Dynamic Test Instrument (Instron Model E1000) decreased apparently and the average bending force calculated by the curve is about 0.193 N (Fig. 3h). The process is just like the relaxation of muscle. After exposing to IR light, the force increases gradually and the increased amplitude

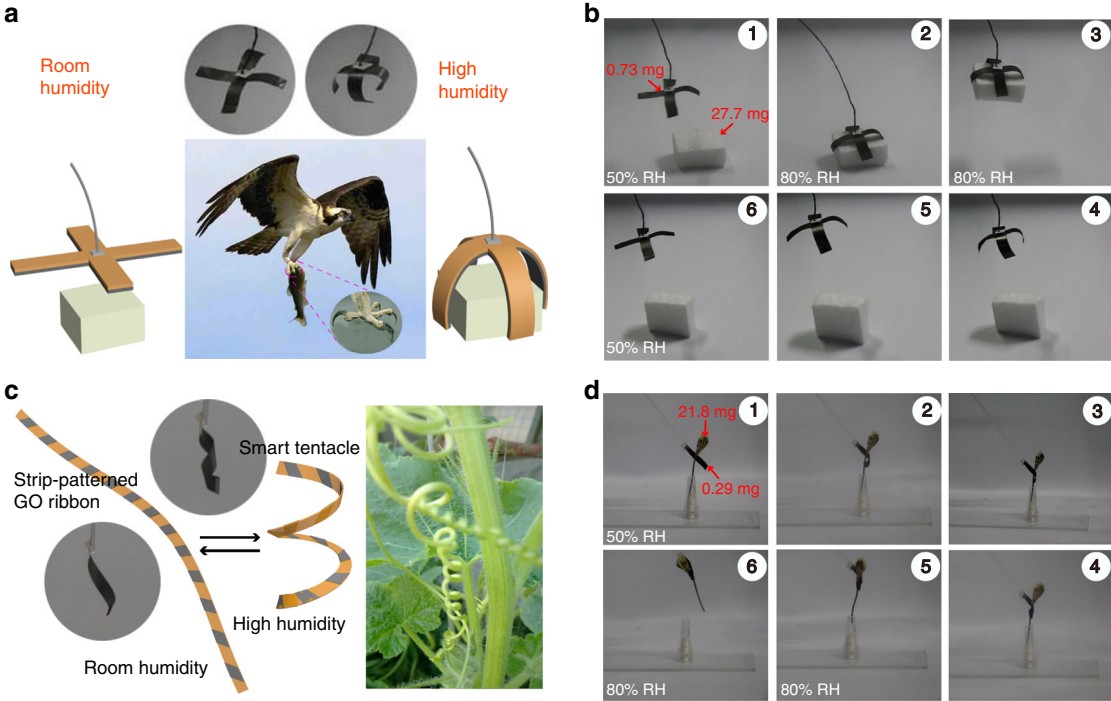

**Fig. 4** Biomimetic gripper with cross and helical structure. **a** Schematic illustration of a smart GO/PPy gripper with a cross structure inspired by the claw of a hawk (middle image). It exhibits a straight state at room humidity (≈50% RH, left images) and a bent state at high humidity (≈80% RH, right images). **b** The process of a foam being picked up and released by a smart GO/PPy gripper; **c** schematic illustration of a smart GO/PPy tendril-inspired by a tendril climber plant (right images). The tendril curves at room humidity (≈50% RH, left images) and curls to a helical structure under high humidity (≈80% RH, middle images); **d** a flower can be removed from a homemade vase by the helical GO/PPy gripper

enlarges with the increasing power density of IR light (Fig. 3i). The process is just like the tension of muscle. The average contractile forces are about 0.092, 0.159 and 0.191 N for light power densities of 0.0423, 0.0831 and 0.126 W cm$^{-2}$, respectively.

**Biomimetic applications.** As mentioned above, GO/PPy bilayer structures possess excellent actuating performance, so programmable and smart actuators were further fabricated to achieve some functions by the designable patterning of PPy on GO films. Inspired by the hawk claw, we designed a smart gripper with a cross structure, which was fabricated by modifying PPy with a cross pattern (Fig. 1b) on the GO film and then cutting it from the GO film. As shown in Fig. 4a, hawks can capture their prey using their claw. The holding power of a hawk's claw is strong enough to allow big animals to be captured and held tightly, although the weight of animal is several times larger than that of the hawk. Similarly, the GO/PPy gripper can bend under the stimulus of humidity and be used to successfully pick up a cuboid polymer foam (14 mm × 14 mm × 7 mm). The weight of the foam (27.7 mg) is 38 times greater than that of GO/PPy gripper (0.73 mg). Figure 4b and Supplementary Movie 3 show the process of the foam being picked up and released. The foam was picked up by the GO/PPy gripper at high humidity and then released when the humidity decreased to a certain value. Furthermore, the GO/PPy gripper recovered its initial state when the humidity decreased to room humidity.

Tendril climber plants can extend and twine around the surrounding support using their helical tendril (the inset in Fig. 4c). We designed a smart tendril by modifying PPy diagonal stripes with 200 μm intervals on a GO film and showed that the tendril can be used as a helical gripper. Figure 4c shows the structure of the GO/PPy tendril at room humidity and high humidity (schematic diagram and photographs). The GO/PPy

tendril remains straight under room humidity and curls to a helical structure under high humidity. We tape one end of the GO/PPy tendril onto a glass capillary and put the GO/PPy tendril onto the stem of a flower at room humidity. The tendril can twine around the stem tightly with increasing humidity, and the flower can be easily removed from our homemade vase, although the weight of flower (21.8 mg) is 75 times larger than that of the GO/PPy tendril (0.29 mg) (Fig. 4d and Supplementary Movie 4). Hence, the GO/PPy tendril can be recognized as helical gripper and may be used in other potential fields.

The movement of the inchworm can be mainly divided into two steps (Fig. 5a): (1) the front feet of the inchworm grip the branch tightly, and the latter part of its body moves forward by curling up its body; (2) the rear feet of the inchworm grip the branch tightly, and the front part of its body moves forward by extending its body. The inchworm will continue moving by repeating this motion. Inspired by the movement of the inchworm, we designed a soft walking robot that can move forward under the interval stimulus of IR light. Figure 5b displayed the schematic diagram of soft walking robot and its movement, the soft walking robot can be divided into two parts: part A and part B. The part A is only the GO film, which has an intersection angle ($\theta = 45°$) with horizontal plane and the angle can be adjusted; the part B is composed of GO and PPy film, which can be recognized as the body of inchworm. Upon the stimuli of IR light, the part B of soft walking robot can bend, and the part A will contact with horizontal plane closely. When the IR light is removed, the part B will recover to the straight state by straightening its body. In this process, the friction ($f$ (a)) between part A and horizontal plane is much larger than the friction ($f$ (b)) between the tip of part B and horizontal plane due to the larger contact area between part A and horizontal plane. Therefore, the soft walking robot will move forward along the direction of $f$ (a), and will move forward for a longer distance by

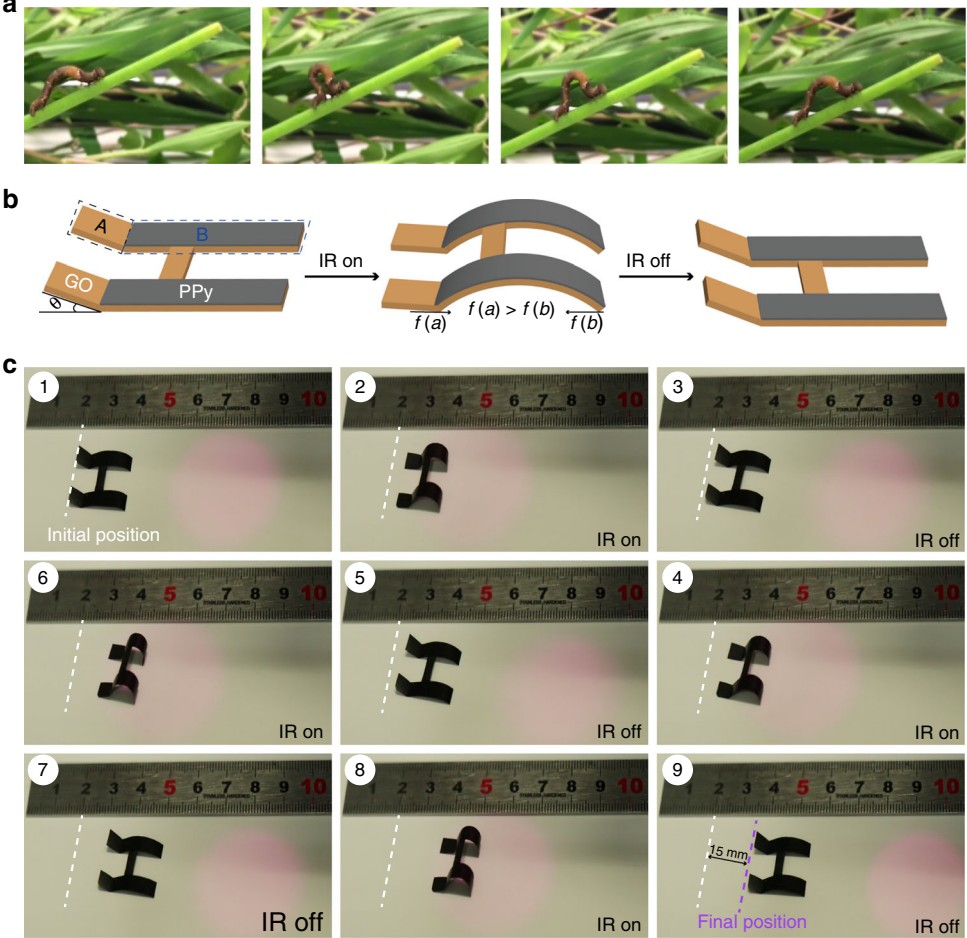

**Fig. 5** Biomimetic soft walking robot. **a** the movement process of the inchworm; **b** the schematic diagram of soft walking robot and its movement; **c** the movement process of the soft walking robot under the interval stimulus of IR light

repeating these processes. The actual movement process of the soft walking robot is shown in Fig. 5c and Supplementary Movie 5. The soft walking robot moves forward by a total of 15 mm after four cycles, so average movement distance can average to 3.75 mm for a single cycle. These results demonstrate that the development of smart walking devices is possible by rational design and precise control over the surrounding environment.

## Discussion

In conclusion, we presented smart and programmable GO/PPy actuators obtained from a simple, time-saving and reliable method for patterning PPy with high precision on a GO film. PANI, PEDOT and calcium alginate hydrogel patterning on GO film, and PPy patterning on GO fibers and foam was also achieved by the same method. The GO/PPy actuator exhibits excellent actuating performance in response to humidity, temperature, and IR stimuli due to the difference of water adsorption or desorption between GO and PPy. Inspired by a hawk's claw and tendril climber plants, we developed smart grippers with cross and helical structures to capture/release target objects. Furthermore, inspired by the movement of the inchworm, we designed a soft walking robot that can move forward for a long distance under the interval stimulus of IR light. Therefore, these programmed actuators can change their structures in a programmable way and we believe that various GO-based smart and programmable actuators will be fabricated for some potential applications via our presented method.

## Methods

**Synthesis of graphene oxide (GO)**. GO was synthesized from graphite powder based on the modified Hummer's method. In detail, 1 g of graphite powder (100 mesh, from Alfa Aesar Reagent Co., Ltd, USA) was mixed with 0.5 g of $NaNO_3$ and 23 mL of $H_2SO_4$ (98%), and the mixture was cooled to 0 °C. Then, 3 g of $KMnO_4$ was added slowly to keep the temperature of the suspension lower than 5 °C and magnetically stirred for 2–3 h. Successively, the reaction system was transferred to a 35 °C water bath and stirred for ~2 h, forming a thick paste. Then, 46 mL of distilled water was added slowly to the solution, and the solution was stirred for 30 min at 95 °C. The mixture was further diluted with 140 mL of distilled water, treated with 2 mL of $H_2O_2$ (30%), washed with 50 mL HCl (1:10) and distilled water 2~3 times and then resuspended in distilled water.

**Synthesis of GO film, GO fiber and GO foam**. For preparation of GO films, 0.5 mL, 1 mL or 1.5 mL of GO solution (15 mg mL$^{-1}$) was poured onto a glass surface, and GO films with different thicknesses were obtained after drying it at ambient temperature for 12 h. The GO fibers were obtained by a wet-spinning method. A 15 mg mL$^{-1}$ GO solution was loaded into a plastic syringe with a spinning nozzle and injected into a rotating coagulation bath (5 wt % $CaCl_2$) at a rate of 30 μL min$^{-1}$. After that, the fiber was transferred into a washing bath (water solution) to wash off the residual salts, and then freeze-drying was conducted to get the GO fiber. The GO foam was obtained by the dip-coating method. A sponge was dipped into GO solution (5 mg mL$^{-1}$) for 1 min, then removed and dried naturally. The process was repeated three times to obtain the resultant GO foam.

**Preparation of PDMS mold by soft lithography**. The standard soft lithography process was used to fabricate the PDMS mold. In brief, a silicon wafer was cleaned and then coated with a SU-8 layer of 150 μm thickness by spinning at 1200 rpm for 60 s. The wafer was prebaked at 65 °C for 15 min and at 95 °C for 120 min and exposed to UV at an energy dose of 700 mJ cm$^{-2}$. The wafer was postbaked at 65 °C for 15 min and 95 °C for 40 min, followed by 3 min of development and hard-baking at 135 °C for 2 h. The PDMS was mixed with a ratio of 10 (base): 1 (curing agent), poured on the SU-8 mold, degassed by a vacuum oven, and cured in

the oven at 70 °C for 4 h. The PDMS mold was subsequently peeled off from the silicon wafer.

**Preparation of a FeCl₃-loaded agarose stamp**. A 4% wt agarose solution was heated to 90 °C, poured onto the PDMS mold treated with oxygen plasma and soaked into the $FeCl_3$ solution for 2 h when the temperature was reduced to room temperature. The $FeCl_3$-loaded agarose stamp was subsequently peeled off from the PDMS mold after the removal of the remaining $FeCl_3$ solution on the surface of the agarose hydrogel.

**Preparation of a CaCl₂-loaded agarose stamp**. A 4% wt agarose solution was heated to 90 °C, poured onto the PDMS mold treated with oxygen plasma and soaked into the 5% $CaCl_2$ solution for 2 h when the temperature was reduced to room temperature. The $CaCl_2$-loaded agarose stamp was subsequently peeled off from the PDMS mold after the removal of the remaining $CaCl_2$ solution on the surface of the agarose hydrogel.

**PPy patterning on GO film, GO fiber and GO foam**. To form PPy patterns on the GO film, a $FeCl_3$-loaded agarose stamp was placed onto the GO film for 20 s and then removed. Next, the pyrrole monomer was dropped onto the GO film and reacted with $FeCl_3$ for 1 min to form PPy patterns. The GO/PPy film was subsequently peeled off from the glass.

PPy patterning on GO fibers and GO foam was conducted with the same procedure as PPy patterning on a GO film

**PANI patterning on GO film**. Similar to the modification of PPy, $FeCl_3$ was transferred onto the GO film by a $FeCl_3$-loaded agarose hydrogel stamp, and $FeCl_3$ was present in only specific patterns replicated from the agarose hydrogel stamp. After that, the mixture of aniline and ethanol (V (aniline): V (ethanol) = 3:2) was dropped onto the GO film and reacted with $FeCl_3$ at the temperature of 60 °C to generate PANI with specific patterns on the GO film.

**PEDOT patterning on GO film**. Similar to the modification of PPy and PANI, $FeCl_3$ was transferred onto the GO film by a $FeCl_3$-loaded agarose hydrogel stamp, and $FeCl_3$ was present in only specific patterns replicated from the agarose hydrogel stamp. After that, the mixture of EDOT and ethanol (V (EDOT): V (ethanol) = 4:1) was dropped onto the GO film and reacted with $FeCl_3$ at the temperature of 60 °C to generate PEDOT with specific patterns on the GO film.

**Calcium alginate hydrogel patterning on GO film**. Similar to the modification of PPy, PANI and PEDOT, $CaCl_2$ was transferred onto the GO film by a $CaCl_2$-loaded agarose hydrogel stamp, and $CaCl_2$ was present in only specific patterns replicated from the agarose hydrogel stamp. After that, 2% sodium alginate was dropped onto the GO film and reacted with $CaCl_2$ to generate calcium alginate with specific patterns on the GO film.

**Testing of actuation behaviors under the stimuli of humidity, heat and IR light**. The actuation behaviors of the GO/PPy ribbon, grippers and soft walking robot were captured by a digital video camera. The humidity was controlled by regulating the water vapor content with an air humidifier, and RH was recorded by a hygrometer. Thermal actuation was measured using a temperature-programmed heating plate. The IR light was from a commercial infrared source, and the distance from the sample to the source was 20 cm.

**Characterization**. Optical images were captured by a super-resolution digital microscope (VHX-2000). The morphology and layered structure of GO/PPy were characterized by field-emission scanning electron microscopy (FESEM, JEM7600F). XPS measurements were performed on a Kratos–Axis spectrometer with monochromatic Al Kα (1486.71 eV) X-ray radiation (15 kV and 10 mA) and a hemispherical electron energy analyzer. XRD patterns were recorded using a diffractometer (X'Pert PRO, Panalytical B.V., Netherlands) equipped with a Cu Kα radiation source ($\lambda$ = 1.5406 Å). Raman spectra were measured on a confocal laser micro-Raman spectrometer (Thermo Fischer DXR, USA) equipped with a He-Ne laser for excitation at 532 nm. FT-IR spectra were recorded using a Bruker spectrometer. The water CAs were measured using an FTA200 contact angle meter, DataPhysics Inc, USA.

## Data availability

All the relevant data used to prepare this manuscript and the Supplementary Information are available from the corresponding author upon reasonable request. The source data underlying Figs. 2f–i and 3a–i and Supplementary Figs. 2, 4C–E, 5C–E, 27, 28, 29 and 31 are provided as a Source Data file.

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

## Acknowledgements

We gratefully acknowledge the financial supports from National Key R&D Program of China (2017YFA0700403, 2016YFF0100801), National Natural Science Foundation of China (21775049, 31700746, 31471257 and 31870856) and the Fundamental Research Funds for the Central Universities (2016YXZD061).

## Author contributions

Y.D., J.W. and B.-F.L. conceived and designed the study, Y.D. did the preliminary experiments, Y.D., J.W., U.D. and B.-F.L. analyzed the experimental data and wrote the paper; B.-F.L. supervised and directed the project. X.K.G., S.S.Y., M.O.O., P.C., X.L., W.D. and F.X. contributed to some experiments and took part in discussions.

## Additional information

**Competing interests:** The authors declare no competing interests.

