## [Peer Review File · Nature Communications]

Reviewers' comments:

Reviewer #1 (Remarks to the Author):

This manuscript reports the development of multi-stimuli responsive bilayer-type film actuators consisting of PPy/GO. The key point of this study is that they developed a method to make a pattern of PPy based on soft-lithography technique with hydrogel. As a result, the authors developed the film actuators whose motion can be programmed based on the patterning of PPy. The authors also demonstrated the responsivenesses of their actuators to temperature, humidity, and IR-irradiation, which indicates the various potentials of their actuators.

At the first glance of this paper, I had rather negative impression on this manuscript, because this paper includes nothing new from the viewpoint of materials science. The authors just combined the materials and techniques, meaning that I could expect all the results without experiments. However, after going through this manuscript several times, I am now positive for accepting this manuscript in Nature Communications due to the following reasons.

- (1) Almost all the reported actuators with patterning were prepared by photo-lithography, while this manuscript realizes the patterning with soft-lithography.
- (2) The patterning method, the authors developed, is very simple but precise, applicable to various materials. In fact, my colleagues also are trying to develop soft actuators with patterning but, as far as I know, their patterns are not as good as ones reported in this manuscript. I think, this manuscript will be helpful to other researchers.
- (3) The demonstrations for various programmed motions of the actuators are interesting and well indicates the potentials of this type of actuators.

Hence, I recommend this manuscript for publications in Nature Communications.

An additional comment of this manuscript is that, I wanted the authors to discuss the relationship between the performances of the actuators and the configurations (thicknesses of two layers and their sizes). For example, relationship between thicknesses and power or response speed. Because the authors can easily prepare the films, in the future, please systematically investigate these fundamental points.

Reviewer #2 (Remarks to the Author):

This manuscript developed GO/PPy bilayer film actuator and demonstrated bendable, helical and movable devices. However, there were many researches about GO actuators containing monolayer or bilayer structures (*Advanced Materials*, 2015, 27, 332; *Acs Nano*, 2012, 6, 8357; *Advanced Functional Materials*, 2017, 27, 1703096; *Advanced Functional Materials*, 2014, 24, 5412; *Scientific reports*, 2015, 5: 9503). The underlying mechanism has demonstrated clearly as water absorption/desorption of GO in previous reports. The overall actuator performance (deformation speed, humidity range, strength and so on) is not outstanding among GO based actuators. Therefore, this manuscript should be transferred to a more specific journal rather than Nature Communications.

1. The actuator has shape-changeability triggered by moisture. However, the performance of this actuator is far from a programmable actuator, which should change the shape or structure in a programmable way.
2. The main factor in this actuator is GO layer, how about changing the other layer into other materials rather than PPy? Which should be simpler ways to construct actuation devices than the method in this manuscript.
3. There are some grammatical mistakes in this manuscript. For example "which are a major barrier to their practical applications with excessive use"
4. For that "the PPy film is not sensitive to changes in RH", the author describes the reason is that the PPy surface is smooth. However, there are not reasonable experiments and explanations,

which is improper and misleading.

5. The performance of inchworm is not good and the actuation cycle is not clear. However, similar actuators have good performance of moving forward for a longer distance without paper assistance in previous studies mentioned above.

Reviewer #3 (Remarks to the Author):

The paper by Dong et al. reported a chemical method for designing GO/PPy bilayer actuator. Multiresponsive actuators are promising smart devices that may find potential applications in soft robots. In this paper, the experiments have been carried out with care and the result is eye-catching, thus, I recommend publication of this paper in Nature Communication. However, before acceptance, the following issues should be addressed.

1. Too many references are cited in the first sentence of the main text.

2. Introduction section (However, apart from photoinduced reactions,...)

Authors introduce many examples of actuators which are fabricated through laser direct writing or other optical preparation method. The intriguing fabrication capability of DLW can prototype the actuator through a single maskless step. However, according to the chemical strategy presented in this manuscript, preparation of PDMS molds and two-step-transferring of the pre-designed patterns onto GO films are needed. What are the advantages of the chemical method compared to photoinduced procedure? The chemical patterning strategy of GO-based actuators has already been reported in Sci Adv (2015, 1, e1500533.). And the recent result published in Adv. Mater (2019, 31, e1806386) shows a graphene spider robot, which was patterned through a simple laser-scribing procedure. Please make a comparison and discuss the advantages of these patterning strategies respectively.

3. Introduction section (Generally, FeCl₃ was transferred onto...)

The detailed preparation steps should be included in the Result or Experiment section, but not in the introduction part.

4. Result (As shown in Figure S2,...)

The paper gives a new strategy of making GO actuators, so different parameters should be investigated and discussed. However, the highest resolution of this method is missing. Authors only demonstrate the difference between the transferred structure and the mold. Moreover, the thickness of the PPy layer should be given. In Fig S5, authors claimed that the PPy film could be adjusted due to different contact time. Actuators with different amounts of PPy should be discussed instead of adjusting GO layer.

5. Result section (IR light (diagram in Figure 1B) .)

I think this sentence is related to Figure 1C, but not 1B.

6. Please add scale bars to all supplementary data.

7. Result (Note that the bending and contraction...)

This chemical pattern strategy should be isotropic, so how to predict and design the bending direction of actuators, what is the internal mechanism, and why are the symmetrical axes not the same under humidity and thermal actuation?

8. Result section (Characteristic wrinkles, which are identical to previously reported results...) and (The characteristic IR peaks of pristine GO agree well with those in the literature....)

References should be cited in these places.

9. Result section (For the PPy side of actuator, a series of peaks...)

Please specify the characteristic peaks of PPy and cite proper references.

10. The unit of curvature should be m⁻¹. However, authors only give the bending angle instead, this should be revised.

11. Fig 3F

The insets in Fig 3F indicated the bending performance under IR light, but the pictures are misleading, please explain why the bending degree of actuator at 32.5°C is smaller than that at 33.0°C, with the initial temperature of 33.2°C.

12. Actuators are investigated for future applications in intelligent machine or robots; please

discuss the mechanical properties of actuators made by this chemical procedure, such as bending forces, energy conversion efficiency and so on.

13. In the result part, authors emphasize that the actuator is multi-responsive (moisture, thermal and IR light), however, in the Biomimetic Application part, the results are only based on moisture response. What contribution does it make to the application of thermal and IR response?

14. For the demonstration of biomimetic inchworm, the transparent square substrate seems to play an important role in creeping, which is misleading for the actuator climbing itself. And one single step is not general for illustrating the walking ability. It is more favored to take more than two steps.

Response to Reviewers' Comments (NCOMMS-19-10183A)

We sincerely thank the reviewers for carefully reviewing our work. In the following, we provide new analyses and new data to address the concerns of the reviewers. The comments of the three reviewers are reproduced below, together with the authors' point-by-point responses on changes made in the revised manuscript to address each reviewer's comments. We feel the manuscript is much improved now by addressing the comments from the reviewers. The comments from referees are reprinted in blue and *italic* Time New Roman. The changes made in the manuscript and supplemental information are marked in red font.

To Reviewer #1:

This manuscript reports the development of multi-stimuli responsive bilayer-type film actuators consisting of PPy/GO. The key point of this study is that they developed a method to make a pattern of PPy based on soft-lithography technique with hydrogel. As a result, the authors developed the film actuators whose motion can be programmed based on the patterning of PPy. The authors also demonstrated the responsivenesses of their actuators to temperature, humidity, and IR-irradiation, which indicates the various potentials of their actuators.

At the first glance of this paper, I had rather negative impression on this manuscript, because this paper includes nothing new from the viewpoint of materials science. The authors just combined the materials and techniques, meaning that I could expect all the results without experiments. However, after going through this manuscript several times, I am now positive for accepting this manuscript in Nature Communications due to the following reasons.

(1) Almost all the reported actuators with patterning were prepared by photo-lithography, while this manuscript realizes the patterning with soft-lithography.

(2) The patterning method, the authors developed, is very simple but precise, applicable to various materials. In fact, my colleagues also are trying to develop soft actuators with patterning but, as far as I know, their patterns are not as good as ones reported in this manuscript. I think, this manuscript will be helpful to other researchers.

(3) The demonstrations for various programmed motions of the actuators are interesting and well indicates the potentials of this type of actuators.

Hence, I recommend this manuscript for publications in Nature Communications.

An additional comment of this manuscript is that, I wanted the authors to discuss the relationship between the performances of the actuators and the configurations (thicknesses of two layers and their sizes). For example, relationship between thicknesses and power or response speed. Because the authors can easily prepare the films, in the future, please systematically investigate these fundamental points.

Response: We greatly appreciate the reviewer's valuable advice. We have further done the experiment and systematically investigated the relationship between the response speed of the actuators and the configurations (thicknesses of two layers and their sizes), as shown in **Supplementary Figure 28** and **Supplementary Figure 29**. **Supplementary Figure 28** exhibits the relationship between the response speed of the actuators (width 2 mm and length 15 mm) and their thicknesses of two layers, a, b, c represent the GO thicknesses of 12.8 μm , 25.2 μm and 37.1 μm respectively, and the PPy thickness increases gradually with the increasing of microcontact time. We can find that the actuators with the GO thickness of 25.2 μm possesses larger response speed at the same microcontact time, this can be explained by the small water-adsorption-induced expansion force for actuator with the GO thickness of 12.8 μm can't bend the actuator quickly, and the large stiffness of actuator with the GO thickness of 37.1 μm cannot bend the actuator flexibly. Furthermore, the actuator with the GO thickness of 25.2 μm and microcontact time of 15 s (**named as GO/PPy (e)**) possesses maximum response speed, which is larger than actuator with the GO thickness of 25.2 μm and microcontact time of 20s (**named as GO/PPy (b)**). This is because the PPy thickness reduces with decreasing microcontact time, so the stiffness of actuator reduces synchronously. However, the actuator GO thickness of 25.2 μm and microcontact time of 10s (**named as GO/PPy (d)**) is smaller in response speed than GO/PPy (b), which is owing to the minor difference of water-adsorption ability between GO and PPy layer at the microcontact time of 10s. Therefore, the proper choice of GO thickness and microcontact time is important for the actuator. **This point has been added to page 14 of the revised manuscript.**

Supplementary Figure 28. The relationship between the response speed of the actuators and the thicknesses of two layers, a, b, c represent the GO thicknesses of 12.8 μm, 25.2 μm and 37.1 μm respectively.

Supplementary Figure 29. The relationship between the response speed of the actuators (GO thickness of 25.2 μm and microcontact time of 20 s) and their sizes.

Supplementary Figure 29 exhibits the relationship between the response speed of the actuators (GO thickness of 25.2 μm and microcontact time of 20 s) and their sizes, we can find clearly that the response speed increases gradually with the increasing actuator length at the same width, and the response speed decreases gradually with the increasing actuator width, which is

ascribed to the larger bending force for the larger actuator length and larger flexural rigidity for the larger width of actuator.

$EI = E_{GO}I_{GO} + E_{PPy}I_{PPy}$ is the flexural rigidity of GO/PPy actuator; I_{GO} and I_{PPy} are the area moments of inertia of GO and PPy layer; E_{GO} and E_{PPy} are constant and the elastic moduli of GO and PPy layer ($E_{GO} = 3073\text{MPa}$; $E_{PPy} = 80\text{MPa}$); b is the width of actuator; h_1 is the thickness of GO layer, h_2 is the thickness of GO/PPy actuator ($h_2 > h_1$).

The relationship of area moments of inertia of GO and the width of actuator can be expressed as:

$$I_{GO} = \frac{2b(h_2^3 - h_1^3)}{3}$$

The relationship of area moments of inertia of PPy and the width of actuator can be expressed as:

$$I_{PPy} = \frac{2bh_1^3}{3}$$

Therefore, the flexural rigidity of GO/PPy actuator can be expressed as

$$EI = E_{GO} \frac{2b(h_2^3 - h_1^3)}{3} + E_{PPy} \frac{2bh_1^3}{3}$$

We can find clearly that the flexural rigidity has no relationship with the length of actuator, which will increase gradually with increasing actuator width, and the actuator is hardly to bend at high flexural rigidity, Therefore, the response speed will decrease gradually with increasing actuator width.

The relationship of bending angle (θ) and the length of actuator (L_{actuator}) can be expressed as:

$$\theta = \frac{F(L)(h_1 + h_2)}{EI}$$

$F(L)$ is proportional to bending force and the value of $F(L)$ and bending force increase with increasing actuator length, so the bending angle increases gradually with the increasing actuator length at the same time and response speed increase correspondingly.

This point has been added to page 14 and page 15 of the revised manuscript.

To Reviewer #2:

This manuscript developed GO/PPy bilayer film actuator and demonstrated bendable, helical and movable devices. However, there were many researches about GO actuators containing monolayer or bilayer structures (Advanced Materials, 2015, 27, 332; Acs Nano, 2012, 6, 8357; Advanced Functional Materials, 2017, 27, 1703096; Advanced Functional Materials, 2014, 24, 5412; Scientific reports, 2015, 5: 9503). The underlying mechanism has demonstrated clearly as water absorption/desorption of GO in previous reports. The overall actuator performance (deformation speed, humidity range, strength and so on) is not outstanding among GO based actuators. Therefore, this manuscript should be transferred to a more specific journal rather than Nature Communications.

Response: We thank the reviewer for these highly valuable comments. The graphene-based materials are ideal choice as fundamental materials to fabricate high-performance actuators due to their fascinating properties, including extraordinary flexibility, low weight density, high mechanical strength and electrical conductivity, unique thermal and optical properties. In particular, graphene oxide (GO), as an important graphene derivative, has been chosen as building blocks for the fabrication of GO-based actuators. The deformation of GO based actuator is indeed caused by the water absorption/desorption of GO, and there are some GO based actuators that have been reported to work by making use of deformation mechanism, which indicate that GO based actuators are very valuable to perform research and explore their applications. It should be noted that the previously reported actuators are almost based on the reduced graphene oxide (RGO)/GO bilayer structure, making use of the difference of property (conductivity, water absorption/desorption) between RGO and GO. However, the difference between RGO and GO is limited, so other materials (polymer, hydrogel, and etc.) should be explored to develop some possible GO based actuators with potentially excellent performances. In this paper, we achieve the modification of PPy on the GO film to fabricate the GO/PPy bilayer actuator, which exhibit outstanding actuating performance. In the meantime, we also achieve the modification of polyaniline (PANI), polyethylene dioxythiophene (PEDOT) and calcium alginate hydrogel by our proposed method, so the development of GO/PANI, GO/PEDOT and hydrogel/GO actuators can also be conducted in the later study. To achieve the programmable structure changes of actuators,

the local modification on GO film is essential. Although PDA have been reported to modify the GO sheet (Advanced Functional Materials, 2014, 24, 5412), which is just involved the total modification. In this paper, the precise and local modification of PPy, PANI, PEDOT and calcium alginate hydrogel can be easily achieved by our proposed method.

The deformation speed of GO/PPy is related to many parameters (the diffusion rate of humidity, the volume of the PMMA cover for providing controlled humidity environment). We conduct the humidity response experiment of actuator in a $0.5 \times 0.5 \times 0.5 \text{ m}^3$ PMMA cover and the diffusion rate of humidity is controlled by the power of air humidifier. The deformation speed of actuator will increase apparently with the increase of the power of air humidifier, but the actuator will be disturbed owing to the large air flow, so the power of air humidifier we used in all humidity response experiment is the optimized result. Furthermore, smaller volume of the PMMA cover will make the diffusion rate of humidity increase, but $0.5 \times 0.5 \times 0.5 \text{ m}^3$ PMMA cover is the minor volume to hold all the experimental instrument we needed in the experiment. Therefore, the deformation speed we measured is the real result in our experimental condition. However, if needed, we can reduce the volume of PMMA cover by using air humidifier and hygrometer with small volume, and increase the power of air humidifier to the maximum power without disturbing of the actuator by adjusting the direction of air flow, so the deformation speed of GO/PPy actuator will increase dramatically. We also conducted the humidity response experiment at maximum power of air humidifier, we can find the actuator curled to a circular ring within 2 s. In general, the deformation speed for a single actuator with the stimuli of humidity is unprecise, which is more suitable to compare the difference of actuators.

The humidity of our experiment is around 50%, the humidity can be increased rapidly and dynamically, so we recorded the humidity response with the humidity from 50% to 70% dynamically. However, the reduction of humidity in our experiment condition is slow and difficult, so the humidity response with the decrease of humidity can't be recorded dynamically. In practice, the actuator also has a response to humidity from 20% to 50%, and GO layer will shrink and the actuator bends to GO layer. The actuator will bend into multiply wound coil when the humidity is larger than 70%. We have measured the curvature of GO/PPy (b) at humidity levels of 20.4%, 31.8%, 43.5% and 50%. As shown in Supplementary Figure 30, the curvature will change from 0 to -0.57 cm^{-1} with the decrease of humidity. The insets in Supplementary Figure 30 show

photographs of the bending GO/PPy (b) at humidity levels of 20.4%, 31.8%, 43.5% and 50%. The bending direction of actuator is opposite when the humidity is under and below 50%, we adopt negative number to express the curvature when the humidity is under 50%. **This point has been added to Page 19 of the ESI.**

Supplementary Figure 30. Actuating performance of GO/PPy (b) at the humidity from 20% to 50%. The insets in Supplementary Figure 30 show photographs of the bending GO/PPy (b) at humidity levels of 20.4%, 31.8%, 43.5% and 50%.

The mechanical strength of actuator can be adjusted on the basis of GO film. The below Figure exhibited the typical stress–strain curves of GO film and GO/PPy actuator. Clearly, the tensile strength of actuator (34.1 MPa) is larger than the GO film (22.0 MPa). We believe that there are more materials that can be modified onto the GO film by our proposed method to fabricate actuators with excellent actuating performance and high tensile strength.

The typical stress-strain curves of GO film and GO/PPy actuator

In conclusion, our work presented smart and programmable GO/PPy actuators obtained from a simple, time-saving and reliable method for PPy patterning with high precision on a GO film. PANI, PEDOT and calcium alginate hydrogel patterning on GO film, and PPy patterning on GO fibers and foam was also achieved by the same method. The GO/PPy actuator exhibits excellent actuating performance in response to humidity, temperature, and IR stimuli. Inspired by a hawk's claw and tendril climber plants, we developed smart grippers with cross and helical structures to capture/release target objects. Furthermore, inspired by the movement of the inchworm, we designed a soft walking robot that can move forward for a long distance under the interval stimulus of IR light. Therefore, these programmed actuators can change their structures in a programmable way and we believe that various GO-based smart and programmable actuators will be fabricated for some potential applications via our presented method.

1. The actuator has shape-changeability triggered by moisture. However, the performance of this actuator is far from a programmable actuator, which should change the shape or structure in a programmable way.

Response: The GO/PPy actuators have shape-changeability under the stimuli of humidity and IR light. The GO/PPy actuators with precise modification of PPy can change their structures in a programmable way. The actuator with the modification of PPy diagonal stripes will curl to a helical structure under high humidity. The H-shaped actuator with the local modification of PPy in the vertical part of 'H' can only bend at the vertical part, which make the actuator can be recognized as a soft walking robot and move forward with the interval stimuli of IR light. Furthermore, the actuators with different shape will change into different structures under the stimuli of humidity and IR light. It should be noted is that the structure changes of GO/PPy actuators can be predicted and controlled by designing the shape and orientation of PPy patterns, and so actuators can change their shape or structure in a programmable way, which is in accordance with previously reported viewpoint. Eugenia Kumacheva et al reported the incorporation of hydrogel diagonal stripes into the network of certain hydrogels with different moduli by lithography to fabricate hydrogel actuators, and these hydrogel actuators can change from planar to cylindrical and conical helical structures (*Nature Communications*, 2013, 4(3):1586; *Advanced Materials*, 2017, 29 (17), 1606111.). Bradley J. Nelson and André R. Studart et al

pre-designed the orientation of magnetic nanoparticles in the hydrogel to fabricate the programmable actuators, and these actuators also can change from planar to cylindrical, conical helical structures and other complex structures (*Nature Communications*, 2016, 7:12263; *Nature Communications*, 2016, 7:13912.). Qu et al reported the GO fiber-type and GO film-type smart actuators with programmable structural changes were successfully prepared by direct laser writing (DLW) on GO fibers and GO films along a pre-designed path (*ACS Nano*, 2016, 10 (10); *Angewandte Chemie International Edition*, 2013, 125 (40), 10676-10680.). In the process of DLW, GO is reduced to reduced GO (RGO), so RGO/GO bilayer structures form on a specific zone of the GO fiber and GO film, which is vital for the formation of actuators due to the apparent difference in expansion and shrinkage between RGO and GO upon adsorption/desorption of water molecules. RGO/GO bilayer actuators with programmable structural changes have also been prepared by UV- and sunlight-irradiation-induced photoreduction of GO films, in which photoreduction of a specific zone is achieved with the assistance of a photomask (*Advanced Functional Materials*, 2015, 25 (28), 4548-4557; *Advanced Materials*, 2015, 27 (2), 332-338.)

We clarified that aspect in the discussion section.

2. *The main factor in this actuator is GO layer, how about changing the other layer into other materials rather than PPy? Which should be simpler ways to construct actuation devices than the method in this manuscript.*

Response: Thank you very much for your kind suggestion. We have achieved the precise modification of PANI, PEDOT and calcium alginate hydrogel onto the GO film by our proposed method. As shown in Supplementary Figure 4, Supplementary Figure 5 and Supplementary Figure 6.

Supplementary Figure 4. The preparation and structural characterization of GO/PANI. (A) Schematic diagram of precise modification of PANI onto the GO film. (B) Optical images of PANI patterns with different sizes and shapes on the GO film; (C) Raman spectra of GO and GO/PANI; (D) FT-IR spectra of pure GO and a GO/PANI. (E) XPS spectra of GO and GO/PANI.

Supplementary Figure 4A showed the schematic diagram of precise modification of PANI onto the GO film. Similar to the modification of PPy, FeCl_3 was transferred onto the GO film by a FeCl_3 -loaded agarose hydrogel stamp, and FeCl_3 was present in only specific patterns replicated from the agarose hydrogel stamp. After that, the mixture of aniline and ethanol (V (aniline): V (ethanol) = 3:2) was dropped onto the GO film and reacted with FeCl_3 to generate PANI with specific patterns on the GO film.

Raman, FT-IR and XPS test were conducted to demonstrate the successful modification of PANI on the GO film. **Supplementary Figure 4B** exhibited the optical images of PANI patterns

with different sizes and shapes on the GO film. We can see clearly that light black PANI patterns are introduced onto the GO film with high precision. Supplementary Figure 4C displayed the Raman spectra of GO and GO/PANI. The spectrum of GO has two dominant peaks at 1350 and 1593 cm^{-1} , corresponding to its D and G bands. Compared to the Raman spectrum of GO, some characteristic Raman bands of PANI appeared in the Raman spectrum of PANI. The Raman band at 443 cm^{-1} is associated with the phenazine-like segment. The band related to out-of-plane C-H deformation of quinonoid ring appears at around 512 cm^{-1} . The 799 and 1160 cm^{-1} bands are assigned to the C-H bending in quinonoid ring and C-H bending deformation in the benzenoid ring. The 1509 cm^{-1} band is associated with C=N stretching vibrations of the quinonoid units. (*Energy & Environmental Science* 6.4 (2013): 1185-1191; *J. Mater. Chem. A*, 2014, 2(41):17489-17494.) The appearance of the above bands suggests the formation of PANI on the surface of GO film. In order to further confirm the formation of PANI on the surface of GO film, the FT-IR analysis was also employed to characterize the GO/PANI. As shown in Supplementary Figure 4D, a group of typical bands of PANI appeared, C=N stretching of the quinonoid ring and C=C stretching of the benzenoid ring at 1601 and 1494 cm^{-1} respectively, C-N stretching of secondary aromatic amines at 1286 cm^{-1} , C-H bendings of the benzenoid ring and the quinonoid ring at 1245 and 1124 cm^{-1} respectively, and C-C stretching of the quinonoid at 794 cm^{-1} . (*Energy & Environmental Science* 6.4 (2013): 1185-1191; *JOURNAL OF PHYSICAL CHEMISTRY C*, 2011, 115(29):14006-14013.) The XPS spectra revealed the surface element compositions of GO and GO/PANI, exhibiting bands at 281.2, 396.4 and 529.8 eV, corresponding to C1s, N1s, and O1s, respectively (Supplementary Figure 4E). The existence of nitrogen element for GO/PANI also demonstrate the formation of PANI on the GO film.

This point has been added to page 7 of the revised manuscript and page 4 of the ESI

Supplementary Figure 5. The preparation and structural characterization of GO/PEDOT. (A) Schematic diagram of precise modification of PEDOT onto the GO film. (B) Optical images of PEDOT patterns with different sizes and shapes on the GO film; (C) Raman spectra of GO and GO/ PEDOT; (D) FT-IR spectra of pure GO and a GO/ PEDOT. (E) XPS spectra of GO and GO/ PEDOT.

Supplementary Figure 5A showed the schematic diagram of precise modification of PEDOT onto the GO film. Similar to the modification of PPy and PANI, FeCl₃ was transferred onto the GO film by a FeCl₃-loaded agarose hydrogel stamp, and FeCl₃ was present in only specific patterns replicated from the agarose hydrogel stamp. After that, the mixture of EDOT and ethanol (V (EDOT): V (ethanol) = 4:1) was dropped onto the GO film and reacted with FeCl₃ to generate PEDOT with specific patterns on the GO film.

Raman, FT-IR and XPS test were conducted to demonstrate the successful modification of

PEDOT on the GO film. **Supplementary Figure 5B** exhibited the optical images of PEDOT patterns with different sizes and shapes on the GO film. We can see clearly that black PEDOT patterns are introduced onto the GO film with high precision. **Supplementary Figure 5C** displayed the Raman spectra of GO and GO/PEDOT. The spectrum of GO has two dominant peaks at 1350 and 1593 cm^{-1} , corresponding to its D and G bands. Compared to the Raman spectrum of GO, some characteristic Raman bands of PEDOT appeared in the Raman spectrum of PEDOT. The Raman bands at 1430 and 1505 cm^{-1} are attributed to the symmetric and asymmetric stretching vibration of the C=C bond in PEDOT, respectively. The band at 1263 cm^{-1} is assigned to the stretching modes of single C-C inter-ring bonds in PEDOT. Other weaker bands at around 990, 856 and 703 cm^{-1} are assigned to C-C asymmetric bond, C-H bending of 2, 3, 5-trisubstituted thiophene and C-S-C bond of PEDOT. (*Electrochimica Acta*, 2013, 108: 118-126; *Macromolecules*, 1999, 32(20): 6807-6812; *Nanoscale Research Letters*, 2007, 2(11): 546.) The appearance of the above bands suggests the formation of PEDOT on the surface of GO film. In order to further confirm the formation of PEDOT on the surface of GO film, the FT-IR analysis was also employed to characterize the GO/PEDOT. As shown in **Supplementary Figure 5D**, a group of typical bands of PEDOT appeared, the feature peaks of PEDOT at about 1518 and 1313 cm^{-1} (C=C and C-C stretching vibrations of the quinoid structure of the thiophene ring) demonstrate the presence of PEDOT. The other bands at about 1182, 1134, and 1047 cm^{-1} are attributed to the C-O-C bond stretching, and the C-S bond in the thiophene ring is proved by the presence of bands at about 975, 829 and 679 cm^{-1} . (*ChemPlusChem*, 2013, 78(3): 227-234; *Carbon*, 2013, 59: 495-502.) The XPS spectra revealed the surface element compositions of GO and GO/PEDOT, exhibiting bands at 161.3, 225.6, 281.2 and 529.8 eV, corresponding to S2p, S2s, C1s and O1s, respectively (**Supplementary Figure 5E**). The existence of sulfur element for GO/PEDOT also demonstrates the formation of PEDOT on the GO film.

This point has been added to page 7 of the revised manuscript and page 5 of the ESI.

Supplementary Figure 6. The preparation and structural characterization of calcium alginate hydrogel/GO. (A) Schematic diagram of precise modification of calcium alginate hydrogel onto the GO film. (B) Optical images of calcium alginate hydrogel patterns with different sizes and shapes on the GO film.

Supplementary Figure 6A showed the schematic diagram of precise modification of calcium alginate onto the GO film. Similar to the modification of PPy, PANI and PEDOT, CaCl₂ was transferred onto the GO film by a CaCl₂-loaded agarose hydrogel stamp, and CaCl₂ was present in only specific patterns replicated from the agarose hydrogel stamp. After that, 2% sodium alginate was dropped onto the GO film and reacted with CaCl₂ to generate calcium alginate with specific patterns on the GO film.

Supplementary Figure 6B exhibited the optical images of calcium alginate patterns with different sizes and shapes on the GO film. We can see clearly that grey white calcium alginate patterns are introduced onto the GO film with high precision.

This point has been added to page 7 of the revised manuscript and page 6 of the ESI.

3. *There are some grammatical mistakes in this manuscript. For example “which are a major barrier to their practical applications with excessive use”*

Response: Thank you very much for your insightful comment, we have polished the grammar of this paper by the professional polishing agency of AMERICAN JOURNAL EXPERTS, and also checked this paper carefully by ourselves for several times. These grammatical mistakes have been modified.

4. *For that “the PPy film is not sensitive to changes in RH”, the author describes the reason is that the PPy surface is smooth. However, there are not reasonable experiments and explanations, which is improper and misleading.*

Response: Thank you very much for pointing out this critical concern, compared to the GO film, the PPy film is not sensitive to changes in RH. As shown in figure below, we can find from the cross-section and surface of PPy film that the PPy film possesses dense structure, which is inconvenient for the transfer of water molecules in the network of PPy film. Furthermore, the hydrophilic functional group is lacking in the PPy film and the interaction between water molecules and PPy film is weak. The polymeric structure of PPy also makes it hydrophobic and the contact angle of PPy film is larger than the GO film (Supplementary Figure 26). However, rich hydrophilic functional groups exist in the GO film, so the interaction between GO film and water molecules is strong. In addition, the unimpeded permeation of water through the GO film has been demonstrated (*Science*, 2012, 335(6067): 442-444.) **This point has been added to page 10 of the revised manuscript.**

5. *The performance of inchworm is not good and the actuation cycle is not clear. However, similar actuators have good performance of moving forward for a longer distance without paper assistance in previous studies mentioned above.*

Response: Thank you for your insightful comment. We have redesigned the structure of actuator to mimic the behavior of inchworm (Figure 5A). The performance of inchworm is greatly improved and the actuation cycle without paper assistance can be found realized. Figure 5B displayed the schematic diagram of soft walking robot and its movement, the soft walking robot can be divided into two parts: part A and part B. The part A is only the GO film, which has an

intersection angle (θ) with horizontal plane; the part B is composed of GO and PPy film, which can be recognized as the body of inchworm. Upon the stimuli of IR light, the part B of soft walking robot can bend, and the part A will contact with horizontal plane closely. When the IR light is removed, the part B will recover to the straight state by straightening its body. In this process, the friction ($f(a)$) between part A and horizontal plane is much larger than the friction ($f(b)$) between the tip of part B and horizontal plane, which is ascribed to the larger contact area between part A and horizontal plane. Therefore, the soft walking robot will move forward along the direction of $f(a)$, and will move forward for a longer distance by repeating these processes. The actual movement process of the soft walking robot is shown in Figure 5C and Supplementary Movie S5. The soft walking robot move forward by a total of 15 mm after four cycles, so average movement distance can average to 3.75 mm for one cycle. These results demonstrate that the development of smart walking devices is possible by rational design and precise control over the surrounding environment. This point has been added to page 18 and page 19 of the revised manuscript.

Figure 5 | Biomimetic soft walking robot. (A) The movement process of the inchworm; (B) the schematic diagram of soft walking robot and its movement; (C) The movement process of the soft walking robot under the interval stimulus of IR light.

To Reviewer #3:

The paper by Dong et al. reported a chemical method for designing GO/PPy bilayer actuator. Multiresponsive actuators are promising smart devices that may find potential applications in soft robots. In this paper, the experiments have been carried out with care and the result is eye-catching, thus, I recommend publication of this paper in Nature Communication. However, before acceptance, the following issues should be addressed.

1. Too many references are cited in the first sentence of the main text.

Response: We thank the reviewer for the kind comments. After careful consideration, we have removed several references cited in the first sentence and reserved some references what we think is essential for this research field.

2. Introduction section (However, apart from photoinduced reactions,...)

Authors introduce many examples of actuators which are fabricated through laser direct writing or other optical preparation method. The intriguing fabrication capability of DLW can prototype the actuator through a single maskless step. However, according to the chemical strategy presented in this manuscript, preparation of PDMS molds and two-step-transferring of the pre-designed patterns onto GO films are needed. What are the advantages of the chemical method compared to photoinduced procedure? The chemical patterning strategy of GO-based actuators has already been reported in Sci Adv (2015, 1, e1500533.). And the recent result published in Adv. Mater (2019, 31, e1806386) shows a graphene spider robot, which was patterned through a simple laser-scribing procedure. Please make a comparison and discuss the advantages of these patterning strategies respectively.

Response: Yes, DLW and lithography are very commonly used for the fabrication of actuators with layered structures, which are amenable to photoinduced reactions, allow designable patterning and create a wide range of proof-of-concept devices, including various self-folding, origami and other complex deformable structures. Particularly, GO fiber-type and GO film-type smart actuators with adjustable structural changes were successfully prepared by DLW on GO fibers and GO films along a predesigned path. In addition, as recent results are published in Adv. Mater (2019, 31, e1806386), a simple laser-scribing procedure also has been used to fabricate

GO-based programmable actuators. In the process of DLW and laser-scribing, GO is reduced to reduced GO (RGO), so RGO/GO bilayer structures form on a specific zone of the GO fiber and GO film, which is vital for the formation of actuators due to the apparent differences in expansion and shrinkage between RGO and GO upon adsorption/desorption of water molecules. However, the difference between RGO and GO is limited, so other materials (polymer, hydrogel, and etc.) should be explored to develop some possible GO based actuators with potentially excellent performances. In addition, apart from photoinduced reactions, local modifications with designable patterning have not been reported. Based on the problem, we proposed the chemical strategy to develop actuators with excellent performances and programmable structure changes. The strategy can allow the precise modification of much more materials produced by chemical reaction onto the GO film, providing more convenience to develop actuators with potentially excellent performance. In this paper, we have achieved the precise modification of PANI, PEDOT, PPy and calcium alginate hydrogel onto the GO film. Although the preparation of PDMS molds and two-step-transferring of the pre-designed patterns onto GO films are needed for chemical strategy, these operations can be achieved easily. Mature technology has been used to fabricate PDMS molds in bulk and the PDMS molds can be used repeatedly. Furthermore, the transfer of pre-designed patterns onto GO films can be achieved manually and high precision still can be ensured.

The modification strategy reported in *Sci Adv* (2015, 1, e1500533.) can achieve the local modification of GO-PDA onto the GO film, which can be recognized as a chemical modification strategy due to the formation of local PDA patterns onto the GO film. However, two-step-filtration have been involved to fabricate the actuator, the second filtration is the key for the patterning and there is no chemical reaction that happens in the patterning process, so the strategy is more like physical patterning strategy. Furthermore, the chemical reaction only happens in the fabrication process of GO-PDA.

In summary, the advantages of photoinduced patterning strategy includes fast speed, allowing designable patterning and precise control, and the advantages of chemical patterning strategy includes high precision, simplicity, allowing designable patterning and more choices for materials. The photoinduced patterning strategy mainly involves photoinduced reaction. The chemical patterning strategy mainly involves chemical reaction. As a result, these patterning strategies can

complement each other, different patterning strategies can be chosen to meet the different requirements of actuators. We have made a comparison between chemical patterning strategy (Sci Adv (2015, 1, e1500533.); and etc.) and photoinduced patterning strategy (Adv. Mater (2019, 31, e1806386); and etc.) in page 5 of the manuscript.

3. Introduction section (Generally, FeCl₃ was transferred onto...)

The detailed preparation steps should be included in the Result or Experiment section, but not in the introduction part.

Response: We thank the reviewer for the kind comments and suggestions. We have deleted the detailed description of preparation steps in the introduction part and added the detailed description of preparation steps in the experiment.

4. Result (As shown in Figure S2,...)

The paper gives a new strategy of making GO actuators, so different parameters should be investigated and discussed. However, the highest resolution of this method is missing. Authors only demonstrate the difference between the transferred structure and the mold. Moreover, the thickness of the PPy layer should be given. In Fig S5, authors claimed that the PPy film could be adjusted due to different contact time. Actuators with different amounts of PPy should be discussed instead of adjusting GO layer.

Response: We thank the reviewer for raising this point. The highest resolution of this method can reach to 60 μ m (Supplementary Figure 3). We measure the thickness of PPy layer by the cross-sectional SEM image of GO/PPy, and the thickness of PPy layer for GO/PPy (a), GO/PPy (b) and GO/PPy (c) is about 15.2 μ m. The thickness of PPy layer will increase with the increase of contact time of FeCl₃-loaded agarose hydrogel stamp onto the GO film (Supplementary Figure 9). As shown in Figure 3B below, the relationship between actuating performance and contact time is also investigated. The thickness of GO film used in the experiment is kept the same and is equal to the thickness of GO layer in GO/PPy (b), and the contact time of 10 s, 15 s and 20 s are chosen, so the corresponding actuators are named as GO/PPy (d), GO/PPy (e) and GO/PPy (b), respectively. Clearly, we can find that the GO/PPy (e) possesses the best actuating performance, which is slightly larger than GO/PPy (b). This can be explained by smaller stiffness for GO/PPy (e). The

PPy thickness reduces with the decreasing microcontact time, so the stiffness of actuator reduces synchronously. The GO/PPy (d) is smaller in actuating performance than GO/PPy (b) and GO/PPy (e), which is owing to the minor difference of water adsorption ability between GO and PPy layer for the GO/PPy (d). Considering the increasing mechanical strength with the increase of PPy thickness and the minor difference of actuating performance between GO/PPy (e) and GO/PPy (b), so we choose GO/PPy (b) for further studies.

This point has been added to page 7 and page 13 of the revised manuscript.

Supplementary Figure 3. The optical image of PPy stripes with 60µm width.

Figure 3B. Actuating performance of GO/PPy actuators under the stimulus of humidity, (GO/PPy (d), GO/PPy (e) and GO/PPy (b) represent GO/PPy actuators with increasing PPy thicknesses and same GO thickness of 25.2 µm.

5. Result section (IR light (diagram in Figure 1B).)

I think this sentence is related to Figure 1C, but not 1B.

Response: Thank you for your insight comment, we have revised the obvious mistake and replaced “IR light (diagram in Figure 1B)” by “IR light (diagram in Figure 1C)” in the revised manuscript.

6. Please add scale bars to all supplementary data.

Response: Scale bars have been added onto all related supplementary Figures.

7. Result (Note that the bending and contraction...)

This chemical pattern strategy should be isotropic, so how to predict and design the bending direction of actuators, what is the internal mechanism, and why are the symmetrical axes not the same under humidity and thermal actuation?

Response: Yes, the chemical pattern strategy is isotropic, and the symmetrical axes are the same for the humidity and IR actuation. We just want to display the obvious structure changes of actuator with different shapes clearly, so we choose different shooting angles to capture the images what we think is optimal for the humidity and IR actuation. This may cause confusion, but the bending and contraction of the actuator are always along the axis of symmetry under the stimuli of humidity and IR light when the actuator has a regular pattern, such as a regular triangle, square, regular pentagon or regular hexagon, and the symmetrical axes are the same for the humidity and IR actuation. The bending direction is mainly determined by the longer symmetrical axes and larger structure change of actuator will happen along the direction, which is the internal mechanism for the bending of actuator. **This point has been added in page 9 of the revised manuscript.**

*8. Result section (Characteristic wrinkles, which are identical to previously reported results...)
and (The characteristic IR peaks of pristine GO agree well with those in the literature....)*

References should be cited in these places.

Response: We truly appreciate the reviewer’s advice. Corresponding references have been cited in these places. (Characteristic wrinkles, which are identical to previously reported results (*ACS nano*, 2016, 10(10): 9529-9535.)); (The characteristic IR peaks of pristine GO agree well with those in the literature (*Energy & Environmental Science*, 2013, 6(4): 1185-1191; *Journal of Materials*

Chemistry A, 2014, 2(13): 4642-4651.)). We can find them in page 9 and Page 12 of the revised manuscript, References [54], [64], [65].

9. Result section (For the PPy side of actuator, a series of peaks...)

Please specify the characteristic peaks of PPy and cite proper references.

Response: We appreciate the reviewer's advice. We have specified the characteristic peaks of PPy in the Raman spectrum of GO/PPy (b) (Figure 2H), and cited some proper references (*Nano Energy*, 2016, 19: 391-400; *Advanced materials*, 2013, 25(4): 591-595.). This point has been added in page 11 and page 12 of the revised manuscript.

Figure 2. (H) Raman spectra of GO and GO/PPy

10. The unit of curvature should be m^{-1} . However, authors only give the bending angle instead, this should be revised.

Response: We thank the reviewer for pointing this out. Yes, many papers have reported to express the curvature by $1/r$ (r is the radius), so we have revised these relevant images.

11. Fig 3F

The insets in Fig 3F indicated the bending performance under IR light, but the pictures are

misleading, please explain why the bending degree of actuator at 32.5 °C is smaller than that at 33.0 °C, with the initial temperature of 33.2 °C.

Response: We thank the reviewer for pointing out this error and have made the correction. The photothermal conversion of GO/PPy (b) is very fast and the temperature of actuator will quickly increase to about 33.0 °C. In the bending process, the change of temperature is very small. However, the temperature difference exist between actual value and observed value measured by infrared thermal imager, which may be misleading. We remeasured the temperature change for several times and found the temperature will increase with the increase in bending degree. **The following is the correct Figure and we replaced Figure 4C in the revised manuscript by it.**

Figure 3. (G) Actuating performance of GO/PPy actuators under the stimulus of IR light (the insets are photographs of the bending GO/PPy ribbon at different times.)

12. Actuators are investigated for future applications in intelligent machine or robots; please discuss the mechanical properties of actuators made by this chemical procedure, such as bending forces, energy conversion efficiency and so on.

Response: We appreciate the reviewer's advice. The bending force generated by GO/PPy (b) under the stimuli of humidity was recorded by All-Electrodynamic Dynamic Test Instrument (Instron Model E1000, England). As shown in **Figure 3H**, the average bending force is about 0.193 N. The contractile force generated by GO/PPy (b) under the stimuli of IR light with different power densities was also measured on the All-Electrodynamic Dynamic Test Instrument.

As shown in **Figure 3I**, the average contractile forces are about 0.092, 0.159 and 0.191 N for light power densities of 0.0423, 0.0831 and 0.126 W cm⁻², respectively. **This point has been added to page 16 and page 17 of the revised manuscript.**

The energy conversion efficiency (η) of a bilayer actuator can be defined as the total elastic energy generated by the GO/PPy (b) actuator divided by the input laser energy (Q_{Laser}). The elastic energy of our actuators can be calculated as follow:

$$Q_{\text{Elastic}} = \frac{[E_2 t_2^2 (3t_1 + t_2) + E_1 t_1^2 (3t_2 + t_1)] [E_1^2 t_1^4 + E_2^2 t_2^4 + 2E_1 E_2 t_1 t_2 (2t_1^2 + 2t_2^4 + 3t_1 t_2)]}{36E_1 E_2 t_1^2 t_2^2 (t_1 + t_2)} \times \left(\frac{1}{r}\right)^2 V_{\text{actuator}}$$

where Q_{Elastic} is the total elastic energy generated by GO/PPy (b) actuator under the stimuli of IR light with power densities (ρ_{Laser}) of 0.0831 W cm⁻². $1/r$ is the curvature (1.40 cm⁻¹), r is the radius of curvature. t_1 and t_2 are the thickness of PPy layer (15.2 μm) and GO layer (37.2 μm) respectively. E_1 and E_2 are the Young's modulus of PPy layer (80 MPa) and GO layer (3.07 GPa) respectively. The Young's modulus of the GO layer is measured by an All-Electrodynamic Dynamic Test Instrument (Instron Model E1000, England). The Young's modulus of the PPy layer has been reported in many paper with the same value (*Materials Science and Engineering: C*, 2008, 28(3):421-428; *Smart Materials and Structures*, 2006, 15(2):243.).

The calculated total elastic energy of GO/PPy (b) actuator is 0.00137 J.

The actuating time (τ) is 3s. A_{actuator} represent the area exposed on the IR light. The input laser energy applied to the actuator can be expressed as:

$$Q_{\text{Laser}} = \rho_{\text{Laser}} \times \tau \times A_{\text{actuator}}$$

The calculated input laser energy of GO/PPy (b) actuator is 0.07479 J.

Hence, the energy conversion efficiency η is given by:

$$\eta = \frac{Q_{\text{Elastic}}}{Q_{\text{Laser}}}$$

The energy conversion efficiency η is then calculated to be 1.832 %.

This point has been added to page 16 of the revised manuscript.

Figure 3. (H) The bending force of GO/PPy (b) actuator recorded during the process of turning on/off the humidifier; (I) The contractile force of GO/PPy (b) actuator recorded during the process of turning on/off IR light with power densities of 0.0423, 0.0831 and 0.126 W cm⁻² (from left to right).

13. In the result part, authors emphasize that the actuator is multi-responsive (moisture, thermal and IR light), however, in the Biomimetic Application part, the results are only based on moisture response. What contribution does it make to the application of thermal and IR response?

Response: We thank the reviewer for pointing out this issue. The temperature surrounding the actuator is hard to control, so the moisture and IR light are more commonly used to propel the actuator in the biomimetic applications. Biomimetic grippers with cross and helical structure have been propelled by moisture to achieve their functions (Figure 4). In the biomimetic application part, we redesigned the structure of actuator to mimic the behavior of inchworm, and the actuator can move forward for a long distance with the interval stimuli of IR light (Figure 5). We consider that different stimuli might fit better to actuate under different environmental conditions.

Figure 4 | Biomimetic gripper with cross and helical structure. (A) Schematic illustration of a smart GO/PPy gripper with a cross structure inspired by the claw of a hawk (middle image). It exhibits a straight state at room humidity ($\approx 50\%$ RH, left images) and a bent state at high humidity ($\approx 80\%$ RH, right images). (B) The process of a foam being picked up and released by a smart GO/PPy gripper; (C) Schematic illustration of a smart GO/PPy “tendrils” inspired by a tendril climber plant (right images). The tendrils curve at room humidity ($\approx 50\%$ RH, left images) and curls to a helical structure under high humidity ($\approx 80\%$ RH, middle images); (D) A flower can be removed from a homemade vase by the helical GO/PPy gripper.

Figure 5 | Biomimetic soft walking robot. (A) The movement process of the inchworm; (B) the schematic diagram of soft walking robot and its movement; (C) The movement process of the soft walking robot under the interval stimulus of IR light.

14. For the demonstration of biomimetic inchworm, the transparent square substrate seems to play an important role in creeping, which is misleading for the actuator climbing itself. And one single step is not general for illustrating the walking ability. It is more favored to take more than two steps.

Response: We appreciate this suggestion from the reviewer. We have redesigned the structure of actuator to mimic the behavior of inchworm (Figure 5A). The performance of inchworm is greatly improved and the actuation cycle without the assistance of transparent square substrate can be found clearly. Figure 5B displayed the schematic diagram of soft walking robot and its movement, the soft walking robot can be divided into two parts: part A and part B. The part A is only the GO film, which has an intersection angle ($\theta = 45^\circ$) with horizontal plane; the part B is composed of GO and PPy film, which can be recognized as the body of inchworm. Upon the stimuli of IR light,

the part B of soft walking robot can bend, and the part A will contact with horizontal plane closely. When the IR light is removed, the part B will recover to the straight state by straightening its body. In this process, the friction ($f(a)$) between part A and horizontal plane is much larger than the friction ($f(b)$) between the tip of part B and horizontal plane, which is ascribed to the larger contact area between part A and horizontal plane. Therefore, the soft walking robot will move forward along the direction of $f(a)$, and will move forward for a longer distance by repeating these processes. The actual movement process of the soft walking robot is shown in Figure 5C and Supplementary Movie S5. The soft walking robot move forward by a total of 15 mm after four cycles, so average movement distance can average to 3.75 mm for one cycle. These results demonstrate that the development of smart walking devices is possible by rational design and precise control over the surrounding environment. This point has been added to page 18 and page 19 of the revised manuscript.

Figure 5 | Biomimetic soft walking robot. (A) The movement process of the inchworm; (B) the schematic diagram of soft walking robot and its movement; (C) The movement process of the soft walking robot under the interval stimulus of IR light.

REVIEWERS' COMMENTS:

Reviewer #1 (Remarks to the Author):

I appreciate the authors' sincere efforts to address my suggestion. I think the revised manuscript is suitable for the publication in Nature Communications without any modifications.

Reviewer #2 (Remarks to the Author):

This referee has not seen any critical importance of science and technique compared with many previous publications by different groups. The current work, although it seems that results presentation is very fancy, has really big advance on neither the design nor the performance. Since there were so many publications about GO actuators containing monolayer or bilayer structures so far as mentioned for the first version, I shall stick to my opinion and do not recommend the publication.

Reviewer #3 (Remarks to the Author):

After carefully evaluation of the revised paper, I find that the authors have suitably addressed all of the comments from the reviewers. I think the paper becomes acceptable. Therefore, I recommend publication of this paper in Nature Communications. However, in the response to Reviewer 2 and Reviewer 3, the authors have emphasized more than 1 time that most GO-based actuators are based on GO/RGO bilayers, and their properties difference is limited. Actually, a recent paper published in Adv. Mater. (DOI: 10.1002/adma.201901585) has proven the fast deformation of a solo GO film under moisture actuation due to the anisotropic structures (called Quantum confined superfluidics channels). Thus, the author should briefly discuss the merits of this new result in the revised manuscript. The use of one material has distinct advantage in long-term stability, whereas the combination of GO with polymers may suffer from poor interlayer adhesion due to the weak interaction.

The paper may be acceptable after addressing this minor point.

To Reviewer #1:

*I appreciate the authors' sincere efforts to address my suggestion.
I think the revised manuscript is suitable for the publication in Nature Communications without any modifications.*

Response: we would like to thank the reviewer for recommending acceptance of our manuscript.

To Reviewer #2

This referee has not seen any critical importance of science and technique compared with many previous publications by different groups. The current work, although it seems that results presentation is very fancy, has really big advance on neither the design nor the performance. Since there were so many publications about GO actuators containing monolayer or bilayer structures so far as mentioned for the first version, I shall stick to my opinion and do not recommend the publication.

Response: we stand by the fact that our work is novel and is a good fit for the Nature Communications readership.

To Reviewer #3

After carefully evaluation of the revised paper, I find that the authors have suitably addressed all of the comments from the reviewers. I think the paper becomes acceptable. Therefore, I recommend publication of this paper in Nature Communications. However, in the response to Reviewer 2 and Reviewer 3, the authors have emphasized more than 1 time that most GO-based actuators are based on GO/RGO bilayers, and their properties difference is limited. Actually, a recent paper published in Adv. Mater.(DOI: 10.1002/adma.201901585) has proven the fast deformation of a solo GO film under moisture actuation due to the anisotropic structures (called Quantum confined superfluidics channels). Thus, the author should briefly discuss the merits of this new result in the revised manuscript. The use of one material has distinct advantage in long-term stability, whereas the combination of GO with polymers may suffer from poor interlayer adhesion due to the weak interaction.

The paper may be acceptable after addressing this minor point.

Response: We would like to thank the reviewer for recommending publication of our manuscript. We also concur with the reviewer that it will be helpful for the readers if we can discuss the merits of this new result in Adv. Mater. (DOI: 10.1002/adma.20190158) in the revised manuscript. We added additional sentences to the discussions and referred to the new paper. This point has been added in page 4 and 5 of the revised manuscript.